# VERIX: Towards Verified Explainability of Deep Neural Networks

**Min Wu**
Department of Computer Science
Stanford University
minwu@cs.stanford.edu

**Haoze Wu**
Department of Computer Science
Stanford University
haozewu@cs.stanford.edu

**Clark Barrett**
Department of Computer Science
Stanford University
barrett@cs.stanford.edu

## Abstract

We present VERIX (VERIfied eXplainability), a system for producing *optimal robust explanations* and generating *counterfactuals* along decision boundaries of machine learning models. We build such explanations and counterfactuals iteratively using constraint solving techniques and a heuristic based on feature-level sensitivity ranking. We evaluate our method on image recognition benchmarks and a real-world scenario of autonomous aircraft taxiing.

## 1 Introduction

Broad deployment of artificial intelligence (AI) systems in safety-critical domains, such as autonomous driving [18] and healthcare [63], necessitates the development of approaches for trustworthy AI. One key ingredient for trustworthiness is *explainability*: the ability for an AI system to communicate the reasons for its behavior in terms that humans can understand.

Early work on explainable AI includes well-known model-agnostic explainers which produce explanations that remain valid for nearby inputs in feature space. In particular, LIME [42] and SHAP [38] learn simple, and thus interpretable, models locally around a given input. Following LIME, work on Anchors [43] attempts to identify a subset of such input explanations that are (almost) sufficient to ensure the corresponding output value. Such approaches can produce explanations efficiently, however, they do not provide any *formal* guarantees and are thus inappropriate for use in high-risk scenarios. For instance, in healthcare, if a diagnosis model used by a dermatologist has an explanation claiming that it depends only on a patient's skin lesions (such as "plaque", "pustule", and "ulcer" in [10]), yet in actuality, patients with similar such lesions but different skin tones ("Fitzpatrick I-II, III-IV, V-VI" [19]) receive dissimilar diagnoses, then the explanation is not only wrong, but may actually mask bias in the model. Another drawback of model-agnostic approaches is that they often depend on access to training data, which may not always be available (perhaps due to privacy concerns). And even if available, distribution shift can compromise the results.

Recent efforts towards *formal* explainable AI [40] aim to compute rigorously defined explanations that can guarantee *soundness*, in the sense that fixing certain input features is sufficient to ensure the invariance of a model's prediction. However, their work only considers *unbounded* perturbations, which may be too course-grained to be useful (for other limitations, see Section 5). To mitigate those drawbacks, [33] bring in two types of *bounded* perturbations, $\epsilon$-ball and $k$-NN box closure, and show how to compute *optimal robust explanations* with respect to these perturbations for natural

37th Conference on Neural Information Processing Systems (NeurIPS 2023).

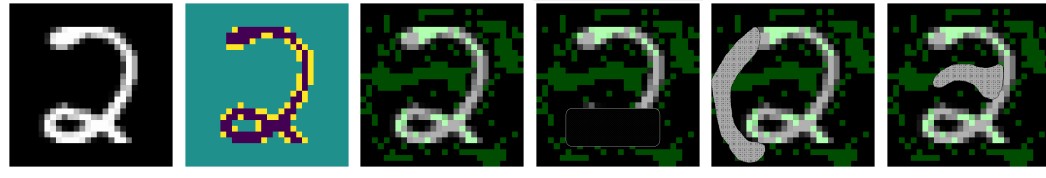

(a) Original "2"  (b) Segmentation  (c) VERIX  (d) "2" into "7"  (e) "2" into "0"  (f) "2" into "3"

Figure 1: Intuition for our VERIX approach: (a) An MNIST handwritten "2"; (b) Segmentation of "2" into 3 partitions; (c) Our VERIX explanation (green pixels) of "2"; (d)(e)(f) Masking white pixels or whitening black pixels may turn "2" into possible counterfactuals.

language processing (NLP) models. $k$-NN box closure essentially chooses a finite set of the $k$ closest tokens for each word in a text sample, so the perturbation space is intrinsically discrete; on the other hand, $\epsilon$-ball perturbations provide a way to handle a continuous word embedding (though the authors of [33] suggest that these may be more cumbersome in NLP applications and focus on $k$-NN box perturbations in their experimental results).

In this paper, we present VERIX (VERIfied eXplainability), a tool for producing *optimal robust explanations* and generating *counterfactuals* along decision boundaries of deep neural networks. Our contributions can be summarized as follows.

- We utilize constraint solving techniques to compute *robust* and *optimal* explanations with provable guarantees against *infinite* and *continuous* perturbations in the input space.
- We bring a *feature-level sensitivity traversal* into the framework to efficiently approximate global optima, which improves scalability for *high-dimensional* inputs and *large* models.
- We note for the first time the relationship between our explanations and *counterfactuals*, and show how to compute such counterfactuals automatically at no additional cost.
- We provide an extensive evaluation on a variety of perception models, including a safety-critical real-world autonomous aircraft taxiing application.

We start by providing intuition for our VERIX approach by analyzing an example explanation in Figure 1. This explanation is generated for a fully-connected model trained on the MNIST dataset. Model-agnostic explainers such as Anchors [43] rely on partitioning an image into a disjoint set of segments and then selecting the most prominent segment(s). Figure 1b shows "2" divided into 3 parts using k-means clustering [37]. Based on this segmentation, the purple and yellow parts would be chosen for the explanation, suggesting that the model largely relies on these segments to make its decision. This also matches our intuition, as a human would immediately identify these pixels as containing information and disregard the background. However, does this mean it is enough to focus on the salient features when explaining a classifier's prediction? Not necessarily. VERIX's explanation is highlighted in green in Figure 1c. It demonstrates that *whatever is prominent is important but what is absent in the background also matters*. We observe that VERIX not only marks those white pixels forming the silhouette of "2" but also includes some background pixels that might affect the prediction if changed. For instance, neglecting the bottom white pixels may lead to a misclassification as a "7"; meanwhile, the classifier also needs to check if the pixels along the left and in the middle are not white to make sure it is not "0" or "3". While Figures 1d, 1e, and 1f are simply illustrative to provide intuition about why different parts of the explanation may be present, we remark that explanations from VERIX are produced automatically and deterministically.

## 2   VERIX: Verified eXplainability

Let $f$ be a neural network and $\mathbf{x}$ a $d$-dimensional input vector of features $\langle \chi^1, \ldots, \chi^d \rangle$. We use $\Theta(\mathbf{x})$, or simply $\Theta$, when the context is clear, to denote its set of feature indices $\{1, \ldots, d\}$. We write $\mathbf{x}^{\mathbf{A}}$ where $\mathbf{A} \subseteq \Theta(\mathbf{x})$ to denote only those features indexed by indices in $\mathbf{A}$. We denote model prediction as $f(\mathbf{x}) = c$, where $c$ is a single quantity in regression or a label among others ($c \in C$) in classification. For the latter, we use $f_c(\mathbf{x})$ to denote the confidence value (pre- or post- softmax) of classifying as $c$, i.e., $f(\mathbf{x}) = \arg\max f_c(\mathbf{x})$. Depending on different application domains, $\mathbf{x}$ can be an image consisting of $d$ pixels as in our case or a text comprising $d$ words as in NLP [33]. In this paper, we focus on perception models. This has the additional benefit that explanations in this context are self-illustrative and thus easier to understand.

## 2.1  Optimal robust explanations

Existing work such as abductive explanations [26], prime implicants [45], and sufficient reasons [11] define a formal explanation as a *minimal* subset of input features that are responsible for a model's decision, in the sense that *any possible* perturbations on the rest features will *never* change prediction. Building on this, [33] introduces *bounded* perturbations and computes "distance-restricted" such explanations for NLP models. Our definition closely follows [33] except: (1) rather than minimizing an arbitrary cost function, we consider the uniform case (i.e., the same cost is assigned to each feature) so as to compute the smallest number of features; (2) we focus on *continuous* $\epsilon$-ball perturbations not *discrete* $k$-NN box closure; (3) we allow for bounded variation (parameterized by $\delta$) in the output to accommodate both classification (set $\delta$ to 0) and regression ($\delta$ could be some pre-defined hyper-parameter quantifying the allowable output change) whilst [33] focuses on sentiment analysis (i.e., binary classification).

**Definition 2.1** (Optimal Robust Explanation). Given a neural network $f$, an input $\mathbf{x}$, a manipulation magnitude $\epsilon$, and a discrepancy $\delta$, a *robust explanation* with respect to norm $p \in \{1, 2, \infty\}$ is a set of input features $\mathbf{x^A}$ such that if $\mathbf{B} = \Theta(\mathbf{x}) \setminus \mathbf{A}$, then

$$\forall\, \mathbf{x^{B'}}. \left\| \mathbf{x^B} - \mathbf{x^{B'}} \right\|_p \leq \epsilon \Rightarrow |f(\mathbf{x}) - f(\mathbf{x'})| \leq \delta, \tag{1}$$

where $\mathbf{x^{B'}}$ is some perturbation on features $\mathbf{x^B}$ and $\mathbf{x'}$ is the input variant combining $\mathbf{x^A}$ and $\mathbf{x^{B'}}$. In particular, we say that the robust explanation $\mathbf{x^A}$ is *optimal* if

$$\forall \chi \in \mathbf{x^A}. \exists\, \mathbf{x^{B'}}, \chi'. \left\| (\mathbf{x^B} \oplus \chi) - (\mathbf{x^{B'}} \oplus \chi') \right\|_p \leq \epsilon \wedge |f(\mathbf{x}) - f(\mathbf{x'})| > \delta, \tag{2}$$

where $\chi'$ is some perturbation of $\chi$ and $\oplus$ denotes concatenation of two features.

We refer to $\mathbf{x^B}$ as the *irrelevant* features. Intuitively, perturbations bounded by $\epsilon$ imposed upon the irrelevant features $\mathbf{x^B}$ will *never* change prediction, as shown by the small blue "+" variants in Figure 2. Moreover, each feature $\chi$ (and their combinations) in the optimal explanation $\mathbf{x^A}$ can be perturbed, together with the irrelevant features, to go beyond the decision boundary, i.e., the orange "+" variants. We mention two special cases: (1) if $\mathbf{x}$ is $\epsilon$-robust, then all features are irrelevant, i.e., $\mathbf{A} = \emptyset$, meaning there is no valid explanation as any $\epsilon$-perturbation does not affect the prediction at all (in other words, a larger $\epsilon$ is required to get a meaningful explanation); (2) if perturbing any feature in input $\mathbf{x}$ can change the prediction, then $\mathbf{A} = \Theta(\mathbf{x})$, meaning the entire input is an explanation.

We remark that our definition of optimality is *local* in that it computes a *minimal* subset of features. An interesting problem would be to find a *globally optimal* explanation, i.e., the smallest (fewest features) among all possible local optima, also known as the cardinality-minimal explanation [26]. Approaches (such as those based on minimum hitting sets [33, 4]) for computing such global optima are often too computationally diffi-

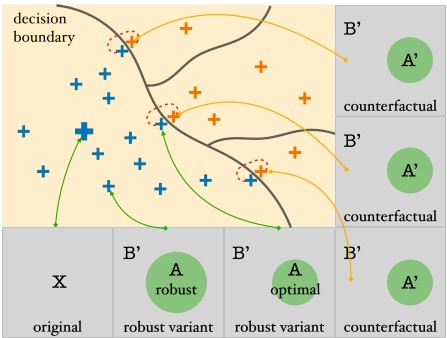

Figure 2: Graphical illustration of VERIX. Each gray square denotes the original input $\mathbf{x}$ (big blue "+") or a variant (smaller "+"). Variants (blue "+") that do not change the explanation $\mathbf{A}$ (green circle) are guaranteed to lie on the same side of the decision boundary. Counterfactuals (orange "+") with perturbed explanations $\mathbf{A'}$ are classified differently.

cult to converge for large models and high-dimensional inputs as in our case. Therefore, we propose a tractable heuristic (Section 3.3) that approximates the ideal and works fairly well in practice.

## 2.2  Counterfactuals along decision boundary

While there are infinitely many variants in the input space, we are particularly interested in those that lie along the decision boundary of a model. Figure 2 shows several pairs of variants (blue and orange "+") connected by red dotted lines. Each pair has the property that the blue variant has the same prediction as the original input $\mathbf{x}$, whereas the orange variant, obtained by further perturbing *one*

Table 1: Evolving index set $\mathbf{A}$ for explanation $\mathbf{x^A}$ and index set $\mathbf{B}$ for irrelevant features $\mathbf{x^B}$ along reasoning result (`True` / `False`) of the CHECK sub-procedure when processing Example 3.1.

| feature | $\mathbf{A}$ | $\mathbf{B}$ | $\mathbf{B'}$ | reasoner | irrelevant features $\mathbf{x^B}$ | explanation $\mathbf{x^A}$ |
|---|---|---|---|---|---|---|
| $\chi^1$ | $\emptyset$ | $\emptyset$ | $\mathbf{B} \cup \{1\}$ | `True` | $\langle \chi^1 \rangle$ | – |
| $\chi^2$ | $\emptyset$ | $\{1\}$ | $\mathbf{B} \cup \{2\}$ | `True` | $\langle \chi^1, \chi^2 \rangle$ | – |
| $\chi^3$ | $\emptyset$ | $\{1, 2\}$ | $\mathbf{B} \cup \{3\}$ | `True` | $\langle \chi^1, \chi^2, \chi^3 \rangle$ | – |
| $\chi^4$ | $\emptyset$ | $\{1, 2, 3\}$ | $\mathbf{B} \cup \{4\}$ | `False` | $\langle \chi^1, \chi^2, \chi^3 \rangle$ | $\langle \chi^4 \rangle$ |
| $\chi^5$ | $\{4\}$ | $\{1, 2, 3\}$ | $\mathbf{B} \cup \{5\}$ | `False` | $\langle \chi^1, \chi^2, \chi^3 \rangle$ | $\langle \chi^4, \chi^5 \rangle$ |
| $\chi^6$ | $\{4, 5\}$ | $\{1, 2, 3\}$ | $\mathbf{B} \cup \{6\}$ | `True` | $\langle \chi^1, \chi^2, \chi^3, \chi^6 \rangle$ | $\langle \chi^4, \chi^5 \rangle$ |
| $\chi^7$ | $\{4, 5\}$ | $\{1, 2, 3, 6\}$ | $\mathbf{B} \cup \{7\}$ | `True` | $\langle \chi^1, \chi^2, \chi^3, \chi^6, \chi^7 \rangle$ | $\langle \chi^4, \chi^5 \rangle$ |
| $\chi^8$ | $\{4, 5\}$ | $\{1, 2, 3, 6, 7\}$ | $\mathbf{B} \cup \{8\}$ | `False` | $\langle \chi^1, \chi^2, \chi^3, \chi^6, \chi^7 \rangle$ | $\langle \chi^4, \chi^5, \chi^8 \rangle$ |
| $\chi^9$ | $\{4, 5, 8\}$ | $\{1, 2, 3, 6, 7\}$ | $\mathbf{B} \cup \{9\}$ | `True` | $\langle \chi^1, \chi^2, \chi^3, \chi^6, \chi^7, \chi^9 \rangle$ | $\langle \chi^4, \chi^5, \chi^8 \rangle$ |

*single* feature $\chi$ in the optimal explanation $\mathbf{x^A}$ (together with the irrelevant features, i.e., $\mathbf{x^{B'}} \oplus \chi'$ in Equation (2), produces a different prediction. We note that these orange variants are essentially *counterfactual* explanations [51]: each is a concrete example of a nearby point in the input space illustrating one way to change the model prediction. We emphasize that our focus in this paper is to compute explanations of the form in Definition 2.1, but it is noteworthy that these counterfactuals are generated automatically and at no additional cost during the computation. In fact, we end up with a distinct counterfactual for each feature in our explanation, as we will see below (see [20, 50] for a comprehensive review of counterfactual explanations and their uses).

## 3 Computing VERIX explanations by constraint solving

Before presenting the VERIX algorithm (Algorithm 1) in detail, we first illustrate it via a simple example.

*Example* 3.1 (VERIX Computation). Suppose $\mathbf{x}$ is an input with 9 features $\langle \chi^1, \ldots, \chi^9 \rangle$ as in Figure 3, and we have classification network $f$, a perturbation magnitude $\epsilon$, and are using $p = \infty$. The outer loop of the algorithm traverses the input features. For simplicity, assume the order of the traversal is from $\chi^1$ to $\chi^9$. Both the explanation index set $\mathbf{A}$ and the irrelevant set $\mathbf{B}$ are initialized to $\emptyset$. At each iteration, VERIX decides whether to add the index $i$ to $\mathbf{A}$ or $\mathbf{B}$. The evolution of the index sets is shown in Table 1. Concretely, when $i = 1$, VERIX formulates a *pre-condition* which specifies that $\chi^1$ can be perturbed by $\epsilon$ while the other features remain unchanged. An automated reasoner is then invoked to check whether the pre-condition logically implies the *post-condition* (in this case, $f(\mathbf{x}) = f(\hat{\mathbf{x}})$, meaning the prediction is the same after perturbation). Suppose the reasoner returns `True`; then, no $\epsilon$-perturbation on $\chi^1$ can alter the prediction. Following Equation (1) of Definition 2.1, we thus add $\chi^1$ to the irrelevant features $\mathbf{x^B}$. Figure 3, top left, shows a visualization of this. VERIX next moves on to $\chi^2$. This time the

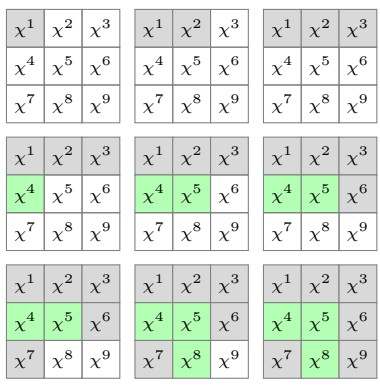

Figure 3: Computing a VERIX explanation by constraint solving for a simple input $\mathbf{x} = \langle \chi^1, \ldots, \chi^9 \rangle$. Green is the *optimal robust explanation* $\mathbf{x^A}$; gray denotes irrelevant features $\mathbf{x^B}$.

precondition allows $\epsilon$-perturbations on both $\chi^1$ and $\chi^2$ while keeping the other features unchanged. The post-condition remains the same. Suppose the reasoner returns `True` again – we then add $\chi^2$ to $\mathbf{x^B}$ (Figure 3, top middle). Following similar steps, we add $\chi^3$ to $\mathbf{x^B}$ (Figure 3, top right). When it comes to $\chi^4$, we allow $\epsilon$-perturbations for $\langle \chi^1, \chi^2, \chi^3, \chi^4 \rangle$ while the other features are fixed. Suppose this time the reasoner returns `False` – there exists a counterexample (this counterexample is the counterfactual for feature $\chi^4$) that violates $f(\mathbf{x}) = f(\hat{\mathbf{x}})$, i.e., the prediction can be different. Then, according to Equation (2) of Definition 2.1, we add $\chi^4$ to the optimal explanation $\mathbf{x^A}$ (shown as green in Figure 3, middle left). The computation continues until all the input features are visited. Eventually, we have $\mathbf{x^A} = \langle \chi^4, \chi^5, \chi^8 \rangle$ (Figure 3, bottom right), which means that, if the features in the explanation are fixed, the model's prediction is invariant to any possible $\epsilon$-perturbation on the other features. Additionally, for each of the features in $\mathbf{x^A}$, we have a counterfactual that demonstrates how that feature can be altered (together with irrelevant features) to change the prediction.

## 3.1 Building optimal robust explanations and counterfactuals iteratively

We now formally describe our VERIX methodology, which exploits an automated reasoning engine for neural network verification as a black-box sub-procedure. We assume the reasoner takes as inputs a network $f$ and a specification

$$\phi_{in}(\hat{\mathbf{x}}) \Rightarrow \phi_{out}(\hat{c}) \tag{3}$$

where $\hat{\mathbf{x}}$ are variables representing the network inputs and $\hat{c}$ are expressions representing the network outputs. $\phi_{in}(\hat{\mathbf{x}})$ and $\phi_{out}(\hat{c})$ are formulas. We use $\hat{\chi}^i$ to denote the variable corresponding to the $i^{th}$ feature. The reasoner checks whether a specification holds on a network.

Inspired by the deletion-based method [8], we propose Algorithm 1, in which the VERIX procedure takes as input a network $f$ and an input $\mathbf{x} = \langle \chi^1, \ldots, \chi^d \rangle$. It outputs an optimal explanation $\mathbf{x}^{\mathbf{A}}$ with respect to perturbation magnitude $\epsilon$, distance metric $p$, and discrepancy $\delta$. It also outputs a set of counterfactuals, one for each feature in $\mathbf{x}^{\mathbf{A}}$. The procedure maintains three sets, $\mathbf{A}$, $\mathbf{B}$, and $\mathbf{C}$, throughout: $\mathbf{A}$ comprises feature indices forming the explanation; $\mathbf{B}$ includes feature indices that can be excluded from the explanation; and $\mathbf{C}$ is the set of counterfactuals. Recall that $\mathbf{x}^{\mathbf{B}}$ denotes the *irrelevant* features (i.e., perturbing $\mathbf{x}^{\mathbf{B}}$ while leaving $\mathbf{x}^{\mathbf{A}}$ unchanged never changes the prediction). To start with, these sets are initialized to $\emptyset$ (Line 2), and the prediction for input $\mathbf{x}$ is recorded as $c$, for which we remark that $c$ may or may not be an *accurate* prediction according to the ground truth – VERIX generates an explanation regardless. Overall, the procedure examines every feature

---

**Algorithm 1** VERIX (VERIfied eXplainability)

**Input**: neural network $f$ and input $\mathbf{x} = \langle \chi^1, \ldots, \chi^d \rangle$
**Parameter**: $\epsilon$-perturbation, norm $p$, and discrepancy $\delta$
**Output**: optimal robust explanation $\mathbf{x}^{\mathbf{A}}$, counterfactuals $\mathbf{C}$ for each $\chi \in \mathbf{x}^{\mathbf{A}}$

1: **function** VERIX($f, \mathbf{x}$)
2:     $\mathbf{A}, \mathbf{B}, \mathbf{C} \mapsto \emptyset, \emptyset, \emptyset$
3:     $c \mapsto f(\mathbf{x})$
4:     $\hat{c} \mapsto f(\hat{\mathbf{x}})$
5:     $\pi \mapsto$ TRAVERSALORDER($\mathbf{x}$)
6:     **for** $i$ in $\pi$ **do**
7:         $\mathbf{B}^+ \mapsto \mathbf{B} \cup \{i\}$
8:         $\phi \mapsto (\left\| \hat{\chi}^{\mathbf{B}^+} - \chi^{\mathbf{B}^+} \right\|_p \leq \epsilon)$
9:         $\phi \mapsto \phi \wedge (\hat{\chi}^{\Theta \backslash \mathbf{B}^+} = \chi^{\Theta \backslash \mathbf{B}^+})$
10:       (HOLD, $m$) $\mapsto$ CHECK($f, \phi \Rightarrow |\hat{c} - c| \leq \delta$)
11:       **if** HOLD **then** $\mathbf{B} \mapsto \mathbf{B}^+$
12:       **else** $\mathbf{A} \mapsto \mathbf{A} \cup \{i\}$ ; $\mathbf{C} \mapsto \mathbf{C} \cup \{m\}$
13:     **return** ($\mathbf{x}^{\mathbf{A}}, \mathbf{C}$)

---

$\chi^i$ in $\mathbf{x}$ according to TRAVERSALORDER (Line 5) to determine whether $i$ can be added to $\mathbf{B}$ or must belong to $\mathbf{A}$. The traversal order can significantly affect the size and shape of the explanation. We propose a heuristic for computing a traversal order that aims to produce small explanations in Section 3.3 (in Example 3.1, a sequential order is used for ease of explanation). For each $i$, we compute $\phi$, a formula that encodes two conditions: (i) the current $\chi^i$ and $\mathbf{x}^{\mathbf{B}}$ are allowed to be perturbed by at most $\epsilon$ (Line 8); and (ii) the rest of the features are fixed (Line 9). The property that we check is that $\phi$ implies $|\hat{c} - c| \leq \delta$ (Line 10), denoting prediction invariance.

An automated reasoning sub-procedure CHECK is deployed to examine whether on network $f$ the specification $\phi \Rightarrow |\hat{c} - c| \leq \delta$ holds (Line 10), i.e., whether perturbing the current $\chi^i$ and irrelevant features while fixing the rest ensures a consistent prediction. It returns (True, $m$) if this is the case (where $m$ is arbitrary) and (False, $m$) if not, where $m$ is a concrete input falsifying the formula. In practice, this CHECK can be instantiated with an off-the-shelf neural network verification tool [46, 41, 31, 56, 21]. If HOLD is True, $i$ is added to the irrelevant set $\mathbf{B}$ (Line 11). Otherwise, $i$ is added to the explanation index set $\mathbf{A}$ (Line 12), which conceptually indicates that $\chi^i$ contributes to the explanation of the prediction (since feature indices in $\mathbf{B}$ have already been proven to not affect prediction). In other words, an $\epsilon$-perturbation that includes the irrelevant features as well as the current $\chi^i$ can breach the decision boundary of $f$. This fact is represented by the counterexample $m$, which represents the counterfactual[1] for $\chi^i$ and is added to the set $\mathbf{C}$ of counterfactuals. The procedure continues until all feature indices in $\mathbf{x}$ are traversed and placed into one of the two disjoint sets $\mathbf{A}$ and $\mathbf{B}$. At the end, $\mathbf{x}^{\mathbf{A}}$ is returned as the optimal explanation and $\mathbf{C}$ is returned as the set of counterfactuals.

---

[1]Properties of counterfactuals such as *actionability* [20] can be addressed by using an appropriate $\epsilon$ to ensure that the counterexamples returned by verifiers will always be from the input data distribution.

## 3.2 Soundness and optimality of explanations

To ensure the VERIX procedure returns a *robust* explanation, we require that CHECK is *sound*, i.e., the solver returns `True` only if the specification actually holds. For the robust explanation to be *optimal*, CHECK also needs to be *complete*, i.e., the solver always returns `True` if the specification holds. We can incorporate various existing reasoners as the CHECK sub-routine. We note that an incomplete reasoner (the solver may return `Unknown`) does *not* undermine the soundness of our approach, though it does affect optimality (the produced explanations may be larger than necessary).

**Lemma 3.2.** *If the* CHECK *sub-procedure is* sound*, then, at the end of each* `for-loop` *iteration (Lines 7–12) in Algorithm 1, the* irrelevant *set of indices* **B** *satisfies*

$$(\left\| \hat{\chi}^{\mathbf{B}} - \chi^{\mathbf{B}} \right\|_p \leq \epsilon) \wedge (\hat{\chi}^{\Theta \setminus \mathbf{B}} = \chi^{\Theta \setminus \mathbf{B}}) \Rightarrow |\hat{c} - c| \leq \delta. \tag{4}$$

Intuitively, any $\epsilon$-perturbation imposed upon all irrelevant features when fixing the others will always keep the prediction consistent, i.e., the infinite number of input variants (indicated with a small blue "+" in Figure 2) will always remain within the decision boundary. We include rigorous proofs for Lemma 3.2 and Theorems 3.3 and 3.4 in Appendix A. Soundness directly follows from Lemma 3.2.

**Theorem 3.3** (Soundness). *If the* CHECK *sub-procedure is* sound*, then the value* $\mathbf{x}^{\mathbf{A}}$ *returned by Algorithm 1 is a* robust *explanation – this satisfies Equation (1) of Definition 2.1.*

**Theorem 3.4** (Optimality). *If the* CHECK *sub-procedure is* sound *and* complete*, then the* robust *explanation* $\mathbf{x}^{\mathbf{A}}$ *returned by Algorithm 1 is* optimal *– this satisfies Equation (2) of Definition 2.1.*

Intuitively, optimality holds because if it is not possible for an $\epsilon$-perturbation on some feature $\chi^i$ in explanation $\mathbf{x}^{\mathbf{A}}$ to change the prediction, then it will be added to the irrelevant features $\mathbf{x}^{\mathbf{B}}$ when feature $\chi^i$ is considered during the execution of Algorithm 1.

**Proposition 3.5** (Complexity). *Given a $d$-dimensional input $\mathbf{x}$ and a network $f$, the* complexity *of computing an optimal robust explanation using the* VERIX *algorithm is $O(d \cdot P(f))$, where $P(f)$ is the cost of checking a specification (of the form in Equation (3)) over $f$.*

An optimal explanation can be achieved from *one* traversal of input features as we are computing the local optima. If $f$ is piecewise-linear, checking a specification over $f$ is NP-complete [30]. We remark that such complexity analysis is closely related to that of the deletion-based method [8]; here we particularly focus on how a network $f$ and a $d$-dimensional input $\mathbf{x}$ would affect the complexity of computing an optimal robust explanation.

## 3.3 Feature-level sensitivity traversal

While Example 3.1 used a simple sequential order for illustration purpose, we introduce a heuristic based on *feature-level sensitivity*, inspired by the occlusion method [64], to produce actual traversals.

**Definition 3.6** (Feature-Level Sensitivity). *Given an input $\mathbf{x} = \langle \chi^1, \ldots, \chi^d \rangle$ and a network $f$, the* feature-level sensitivity *(in classification for a label $c$ or in regression for a single quantity) for a feature $\chi^i$ with respect to a transformation $\mathcal{T}$ is*

$$\mathsf{sensitivity}(\chi^i) = f_{(c)}(\mathbf{x}) - f_{(c)}(\mathbf{x}'), \tag{5}$$

*where $\mathbf{x}'$ is $\mathbf{x}$ with $\chi^i$ replaced by $\mathcal{T}(\chi^i)$.*

Typical transformations include *deletion* ($\mathcal{T}(\chi) = 0$) and *reversal* ($\mathcal{T}(\chi) = \overline{\chi} - \chi$, where $\overline{\chi}$ is the upper bound for feature $\chi$). Intuitively, we measure how sensitive (in terms of an increase or decrease) a model's confidence is to each individual feature. Given sensitivity values with respect to some transformation, we rank the feature indices into a traversal order from least to most sensitive.

## 4 Experimental results

We have implemented the VERIX algorithm in Python, using the Marabou neural network verification tool [31] to implement the CHECK sub-procedure of Algorithm 1 (Line 10). The VERIX code is available at `https://github.com/NeuralNetworkVerification/VeriX`. We trained fully-connected and convolutional networks on the MNIST [34], GTSRB [47], and TaxiNet [29] datasets for classification and regression tasks. Model specifications are in Appendix D. Experiments were performed on a workstation equipped with AMD Ryzen 7 5700G CPUs running Fedora 37. We set a time limit of 300 seconds for each CHECK call.

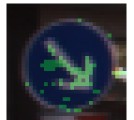 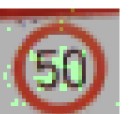 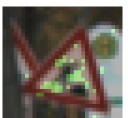 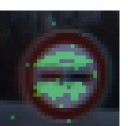 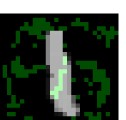 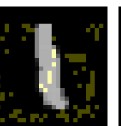 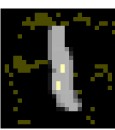 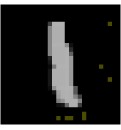

(a) "keep right, 50 mph, road work, no passing"  (b) Handwritten digit "1", not "8", not "5", not "2"

Figure 4: Optimal robust explanations (green) from VERIX on GTSRB (left) and MNIST (right) images. (b) Pixels in yellow are those in the explanation to rule out different counterfactuals.

## 4.1  Example explanations for image recognition benchmarks

Figure 4 shows examples of VERIX explanations for GTSRB and MNIST images. The convolutional model trained on GTSRB and fully-connected model on MNIST are in Appendix D, Tables 8 and 6. Aligning with our intuition, VERIX can distinguish the traffic signs (no matter a circle, a triangle, or a square in Figure 6a) from their surroundings well; the explanations focus on the actual contents within the signs, e.g., the right arrow denoting "keep right" and the number $50$ as in "50 mph". Interestingly, for traffic signs consisting of irregular dark shapes on a white background such as "road work" and "no passing", VERIX discovers that the white background contains the essential features. We observe that MNIST explanations for the fully-connected model are in general more scattered around the background because the network relies on the non-existence of white pixels to rule out different counterfactuals (Figure 4b shows which pixels in the explanation have associated counterfactuals with predictions of "8", "5", and "2", respectively), whereas GTSRB explanations for the convolutional model can safely disregard the surrounding pixels outside the traffic signs.

## 4.2  Visualization of the effect of varying perturbation magnitude $\epsilon$

A key parameter of VERIX is the perturbation magnitude $\epsilon$. When $\epsilon$ is varied, the irrelevant features change accordingly. Figure 5 visualizes this, showing how the irrelevant features change when $\epsilon$ is tightened from $100\%$ to $10\%$ and further to $5\%$. As $\epsilon$ decreases, more pixels become irrelevant. Intuitively, the VERIX explanation helps reveal how the network classifies this image as "0". The deep blue pixels are those that are irrelevant with $\epsilon = 100\%$. Light blue pixels are more sensitive, allowing perturbations of only $10\%$. The light yellow pixels represent $5\%$, and bright yellow are pixels that cannot even be perturbed $5\%$ without changing the prediction. The resulting pattern is roughly consistent with our intuition, as the shape of the "0" can be seen embedded in the explanation.

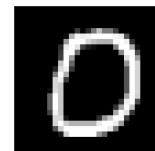 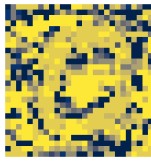 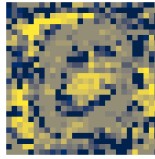

(a) MNIST "0"  (b) $\epsilon : 1 \rightarrow 0.1$  (c) $\epsilon : 1 \rightarrow 0.05$

Figure 5: Expansion of the irrelevant pixels when perturbation magnitude $\epsilon$ decreases from $100\%$ to $10\%$ and further to $5\%$ (from deep blue to light yellow). Each brighter color denotes the pixels added when moving to the next smaller $\epsilon$, e.g., $100\%, 90\%, 80\%$ and so on.

We remark that determining a suitable magnitude $\epsilon$ is non-trivial because if $\epsilon$ is too loose, explanations may be too conservative, allowing very few pixels to change. On the other hand, if $\epsilon$ is too small, nearly the whole set of pixels could become irrelevant. For instance, in Figure 5, if we set $\epsilon$ to $1\%$ then all pixels become irrelevant – the classifier's prediction is robust to $1\%$-perturbations. The "color map" we propose makes it possible to visualize not only the explanation but also how it varies with $\epsilon$. The user then has the freedom to pick a specific $\epsilon$ depending on their application.

## 4.3  Sensitivity vs. random traversal to generate explanations

To show the advantage of the *feature-level sensitivity* traversal, Figure 6 compares VERIX explanations using sensitivity-based and random traversals. Sensitivity, as shown in the heatmaps of Figures 6a and 6b, prioritizes pixels that have more influence on the network's decision, whereas a random ranking is simply a shuffling of all the pixels. We mention that, to compute the sensitivity, we used pixel deletion $\mathcal{T}(\chi) = 0$ for GTSRB and reversal $\mathcal{T}(\chi) = \overline{\chi} - \chi$ for MNIST (deleting background pixels of MNIST images may have little effect as they often have zero values). We observe that the sensitivity traversal generates much more sensible explanations. In Figure 6c, we

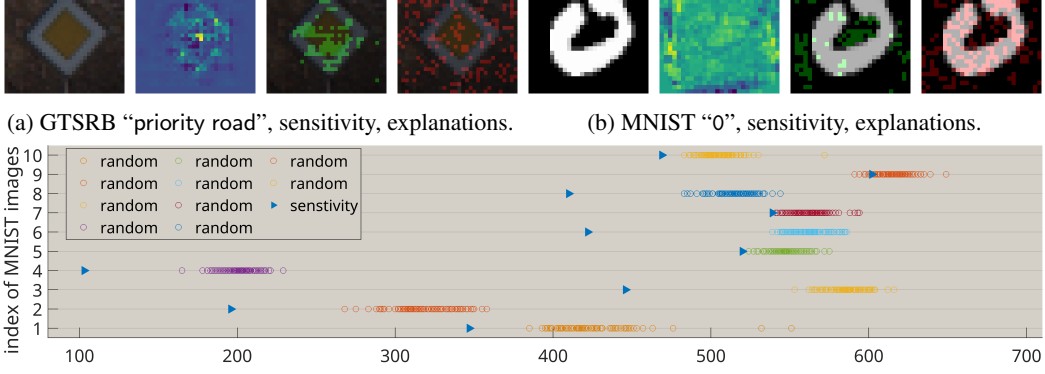

(a) GTSRB "priority road", sensitivity, explanations.     (b) MNIST "0", sensitivity, explanations.

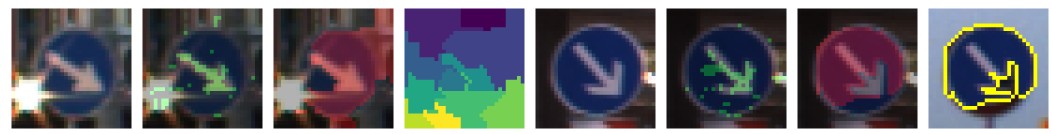

(c) Sensitivity vs. random traversals in explanation size (number of pixels).

Figure 6: Comparing VERIX explanations from *sensitivity* and random traversals. (a)(b) 1st column: the original image; 2nd column: sensitivity heatmap; 3rd and 4th columns: explanations from sensitivity (green) and random (red) traversals. (c) Each blue triangle is a unique explanation size from sensitivity ranking, and each set of circles shows explanation sizes from 100 random rankings.

(a) Original, VERIX, Anchors, segmentation     (b) Original, VERIX, Anchors, misclassified as "yield"

Figure 7: Comparing VERIX (green) to Anchors (red) on two different versions of a "keep right" traffic sign from the GTSRB dataset, one with strong light in the background and one without.

compare explanation sizes for the first 10 images (to avoid potential selection bias) of the MNIST test set. For each image, we show 100 explanations from random traversal compared to the deterministic explanation from sensitivity traversal. We observe that the latter is almost always smaller, often significantly so, suggesting that sensitivity-based traversals are a reasonable heuristic for attempting to approach globally optimal explanations.

## 4.4  VERIX vs. existing approaches

We compare VERIX with Anchors [43]. Figure 7 shows both approaches applied to two different "keep right" traffic signs. Anchors performs image segmentation and selects a set of the segments as the explanation, making its explanations heavily dependent on the quality of the segmentation. For instance, distractions such as strong light in the background may compromise the segments (Figure 7a, last column) thus resulting in less-than-ideal explanations, e.g., the top right region of the anchor (red) is outside the actual traffic sign. Instead, VERIX utilizes the model to compute the sensitivity traversal, often leading to more reasonable explanations. Anchors is also not designed to provide *formal* guarantees. In fact, replacing the background of an anchor explanation – used by the original paper [43] to justify "almost" guarantee – can change the classification. For example, the last column of Figure 7b is classified as "yield" with confidence $99.92\%$. We conducted a quantitative evaluation on the robustness of explanations, as shown in Table 2 under "robust wrt # perturbations". When quantifying the robustness of Anchors, we generate 100 explanations for 100 MNIST and 100 GTSRB images separately, and impose 10, 100, 500, 1000 perturbations on each explanation by overlapping it with other images in the same dataset. If the prediction remains unchanged, then the explanation is robust against these perturbations. We notice that the robustness of Anchors decreases quickly when the number of perturbations increases, and when imposing 1000 perturbations per image, only 37 MNIST and 15 GTSRB explanations out of 100 are robust. This was not done for LIME, since it would be unfair to LIME as it does not have robustness as a primary goal. In contrast, VERIX provides provable robustness guarantees against any $\epsilon$-perturbations in the input space.

Two key metrics for evaluating the quality of an explanation are the *size* and the *generation time*. Table 2 shows that overall, VERIX produces much smaller explanations than Anchors and LIME but

Table 2: VERIX vs. existing approaches, showing average explanation *size* (number of pixels), generation *time* (seconds), and an empirical evaluation of the *robustness* of the produced explanations against perturbations. In VERIX, $\epsilon$ is set to $5\%$ for MNIST and $0.5\%$ for GTSRB.

| | | \multicolumn{6}{MNIST} | | | | | \multicolumn{6}{GTSRB} | | | |
|---|---|---|---|---|---|---|---|---|---|---|---|---|---|
| | | size | time | \multicolumn{4}{robust wrt # perturbations} | | | | size | time | \multicolumn{4}{robust wrt # perturbations} | | | |
| **VERIX** | **sensitivity** | **180.6** | **174.77** | \multicolumn{4}{**100**%} | | | | **357.0** | **853.91** | \multicolumn{4}{**100**%} | | | |
| | random | 294.2 | 157.47 | \multicolumn{4}{100%} | | | | 383.5 | 814.18 | \multicolumn{4}{100%} | | | |
| \multicolumn{2}{Anchors} | | 494.9 | 13.46 | 10
85% | 100
52% | 500
40% | 1000
37% | 557.7 | 26.15 | 10
72% | 100
23% | 500
16% | 1000
15% |
| \multicolumn{2}{LIME} | | 432.8 | 9.03 | \multicolumn{4}{–} | | | | 452.9 | 7.85 | \multicolumn{4}{–} | | | |

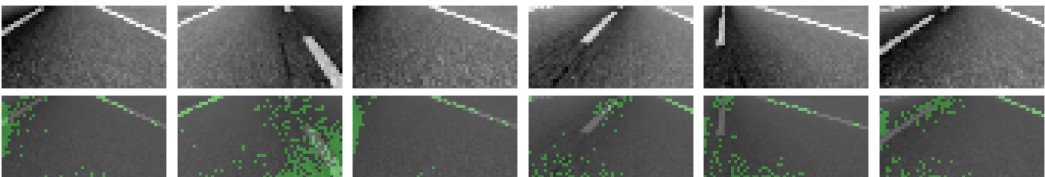

(a) 3.94 (3.91) m  (b) 5.09 (4.97) m  (c) 3.32 (3.38) m  (d) 1.75 (1.55) m  (e) 1.58 (1.60) m  (f) 3.88 (4.07) m

Figure 8: VERIX applied to the TaxiNet dataset – each column includes a sampled camera view (top), its VERIX explanation (bottom), and the cross-track estimate of the form "actual (estimate) meters".

takes much longer (our limitation) to perform the computation necessary to ensure formal guarantees – this thus provides a trade-off between time and quality. Table 2 also shows that sensitivity traversal produces significantly smaller sizes than its random counterpart with only a modest overhead in time.

## 4.5 Deployment in vision-based autonomous aircraft taxiing

We also applied VERIX to the real-world safety-critical aircraft taxiing scenario [29] shown in Figure 9. The vision-based autonomous taxiing system needs to make sure the aircraft stays on the taxiway utilizing only pictures taken from the camera on the right wing. The task is to evaluate the cross-track position of the aircraft so that a controller can adjust its position accordingly. To achieve this, a regression model is used that takes a picture as input and produces an estimate of the current position. A preprocessing step crops out the sky and aircraft nose, keeping the crucial taxiway region (in the red box). This is then downsampled into a gray-scale image of size $27 \times 54$ pixels. We label each image with its corresponding lateral distance to the runway centerline together with the taxiway heading angle. We trained a fully-connected regression network on this dataset, referred to as the TaxiNet model (Appendix D.3, Table 10), to predict the aircraft's cross-track distance.

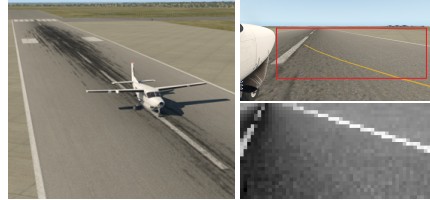

Figure 9: An autonomous aircraft taxiing scenario [29]. Pictures taken from the camera fixed on the right wing are cropped (red box) and downsampled.

Figure 8 exhibits VERIX applied to the TaxiNet dataset, including a variety of taxiway images with different heading angles and number of lanes. For each taxiway, we show its VERIX explanation accompanied by the cross-track estimate. We observe that the model is capable of detecting the more remote line – its contour is clearly marked in green. Meanwhile, the model is mainly focused on the centerline (especially in Figures 8b, 8d, 8e, and 8f), which makes sense as it needs to measure how far the aircraft has deviated from the center. Interestingly, while we intuitively might assume that the model would focus on the white lanes and discard the rest, VERIX shows that the bottom middle region is also crucial to the explanation (e.g., as shown in Figures 8a and 8c). This is because the model must take into account the presence and absence of the centerline. This is in fact in consistent with our observations about the black background in MNIST images (Figure 1). We used $\epsilon = 5\%$ for these explanations, which suggests that for modest perturbations (e.g., brightness change due to different weather conditions) the predicted cross-track estimate will remain within an acceptable discrepancy, and taxiing will not be compromised.

Table 3: Complete vs. *incomplete* verification, showing average explanation *size* (number of pixels) and generation *time* (seconds). As in Table 2, $\epsilon$ is set to $5\%$ for MNIST and $0.5\%$ for GTSRB.

| | | MNIST | | | | GTSRB | | | |
| | | complete | | **incomplete** | | complete | | **incomplete** | |
| | | size | time | size | time | size | time | size | time |
|---|---|---|---|---|---|---|---|---|---|
| VERIX | sensitivity | 180.6 | 174.77 | **330.8** | **106.11** | 357.0 | 853.91 | **385.2** | **858.39** |
| | random | 294.2 | 157.47 | **467.4** | **100.19** | 383.5 | 814.18 | **400.2** | **816.38** |

### 4.6 Using sound but incomplete analysis as the CHECK sub-procedure

We also evaluate using sound but *incomplete* analysis, such as DeepPoly [46], as the CHECK sub-procedure of Algorithm 1. In Table 3, we report the quantitative comparison.

We observe that, in general, explanations are *larger* when using incomplete analysis, but have *shorter* generation times. In other words, there is a trade-off between complete and incomplete analyses with respect to explanation size and generation time. This makes sense as the DeepPoly analysis deploys bound propagation without complete search, so when the verifier returns `Unknown` on a certain feature (Line 10 of Algorithm 1), it is *unknown* whether perturbing this feature (together with the current irrelevant set $\mathbf{B}$) can or cannot change the model's prediction. Therefore, such features are put into the explanation set $\mathbf{A}$, as only $\epsilon$-robust features are put into the irrelevant set $\mathbf{B}$. This results in larger explanations. We re-emphasize that such explanations are no longer locally minimal, but they are still *sound* as $\epsilon$-perturbations on the irrelevant features will definitely not alter the prediction. Another observation is that on MNIST, the discrepancy between explanations (size and time) from complete and incomplete verifiers is more pronounced than the discrepancy on GTSRB. This is because here we set $\epsilon$ to $5\%$ for MNIST and to $0.5\%$ for GTSRB. So when generating explanations on MNIST, complete search is often required for a precise answer due to the larger $\epsilon$ value, whereas on GTSRB, bound propagation is often enough due to the small $\epsilon$, and thus complete search is not always needed.

Due to space limitations, additional analyses of *runtime performance* and *scalability* for both complete and incomplete verifiers are included in Appendix B.

## 5 Related work

Earlier work on *formal* explanations [40] has the following limitations. First, in terms of *scalability*, they can only handle simple machine learning models such as naive Bayes classifiers [39], random forests [28, 6], decision trees [27], and boosted trees [24, 25]. In particular, [26] addresses networks but with very simple structure (e.g., one hidden layer of 15 or 20 neurons to distinguish two MNIST digits). In contrast, VERIX works with models applicable to real-world safety-critical scenarios. Second, the *size* of explanations can be unnecessarily large. As a workaround, approximate explanations [52, 53] are proposed as a generalization to provide probabilistic (thus compromised) guarantees of prediction invariance. VERIX, by using feature-level sensitivity ranking, produces reasonably-sized explanations with rigorous guarantees. Third, these formal explanations allow *any possible input* in feature space, which is not necessary or realistic. For this, [33] brings in *bounded* perturbations for NLP models but their $k$-NN box closure essentially chooses a finite set of the $k$ closest tokens for each word so the perturbation space is intrinsically discrete. In other words, their method will not scale to our high-dimensional image inputs with infinite perturbations. One the other hand, those model-agnostic feature selection methods such as Anchors [43] and LIME [42] can work with large networks, however, they cannot provide strong guarantees like our approach can. A brief survey of work in the general area of neural network verification appears in Appendix C.

## 6 Conclusions and future work

We have presented the VERIX framework for computing optimal robust explanations and counterfactuals along the decision boundary. Our approach provides provable guarantees against infinite and continuous perturbations. A possible future direction is generalizing to other crucial properties such as *fairness*. Recall the diagnosis model in Section 1; our approach can discover potential bias (if it exists) by including skin tones ("Fitzpatrick I-II, III-IV, V-VI" [19]) in the produced explanation; then, a dermatologist could better interpret whether the model is producing a fair and unbiased result.

## Acknowledgments

This work was supported in part by NSF grant 2211505, the Stanford Center for AI Safety, and the Stanford Center for Automated Reasoning.

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

# A   Proofs for Section 3.2

We present rigorous proofs for Lemma 3.2, Theorems 3.3 and 3.4 in Section 3.2, justifying the *soundness* and *optimality* of our VERIX approach. For better readability, we repeat each lemma and theorem before their corresponding proofs.

## A.1   Proof for Lemma 3.2

*Lemma 3.2.* If the CHECK sub-procedure is *sound*, then, at the end of each `for-loop` iteration (Lines 7–12) in Algorithm 1, the *irrelevant* set of indices $\mathbf{B}$ satisfies

$$(\left\|\hat{\chi}^{\mathbf{B}} - \chi^{\mathbf{B}}\right\|_p \leq \epsilon) \wedge (\hat{\chi}^{\Theta \setminus \mathbf{B}} = \chi^{\Theta \setminus \mathbf{B}}) \Rightarrow |\hat{c} - c| \leq \delta. \tag{6}$$

*Proof.* Recall that the sub-procedure CHECK is *sound* means the deployed automated reasoner returns `True` only if the specification actually holds. That is, from Line 10 we have

$$\phi \Rightarrow |\hat{c} - c| \leq \delta$$

holds on network $f$. Simultaneously, from Lines 8 and 9 we know that, to check the current feature $\chi^i$ of the traversing order $\pi$, the pre-condition $\phi$ contains

$$\phi \mapsto (\left\|\hat{\chi}^{\mathbf{B}^+} - \chi^{\mathbf{B}^+}\right\|_p \leq \epsilon) \wedge (\hat{\chi}^{\Theta \setminus \mathbf{B}^+} = \chi^{\Theta \setminus \mathbf{B}^+}).$$

Specifically, we prove this through induction on the number of iteration $i$. When $i$ is 0, pre-condition $\phi$ is initialized as $\top$ and the specification holds trivially. In the inductive case, suppose CHECK returns `False`, then the set $\mathbf{B}$ is unchanged as in Line 12. Otherwise, if CHECK returns `True`, which makes HOLD become `True`, then the current feature index $i$ is added into the irrelevant set of feature indices $\mathbf{B}$ as in Line 11, with such satisfying specification

$$(\left\|\hat{\chi}^{\mathbf{B}^+} - \chi^{\mathbf{B}^+}\right\|_p \leq \epsilon) \wedge (\hat{\chi}^{\Theta \setminus \mathbf{B}^+} = \chi^{\Theta \setminus \mathbf{B}^+}) \Rightarrow |\hat{c} - c| \leq \delta.$$

As the iteration proceeds, each time CHECK returns `True`, the irrelevant set $\mathbf{B}$ is augmented with the current feature index $i$, and the specification always holds as it is explicitly checked by the CHECK reasoner. □

## A.2   Proof for Theorem 3.3

*Theorem 3.3* (Soundness). If the CHECK sub-procedure is *sound*, then the value $\mathbf{x}^{\mathbf{A}}$ returned by Algorithm 1 is a *robust* explanation – this satisfies Equation (1) of Definition 2.1.

*Proof.* The `for-loop` from Line 6 indicates that Algorithm 1 goes through every each feature $\mathbf{x}^i$ in input $\mathbf{x}$ by traversing the set of indices $\Theta(\mathbf{x})$. Line 5 means that $\pi$ is one such instance of ordered traversal. When the iteration ends, all the indices in $\Theta(\mathbf{x})$ are either put into the irrelevant set of indices by $\mathbf{B} \mapsto \mathbf{B}^+$ as in Line 11 or the explanation index set by $\mathbf{A} \mapsto \mathbf{A} \cup \{i\}$ as in Line 12. That is, $\mathbf{A}$ and $\mathbf{B}$ are two disjoint index sets forming $\Theta(\mathbf{x})$; in other words, $\mathbf{B} = \Theta(\mathbf{x}) \setminus \mathbf{A}$. Therefore, combined with Lemma 3.2, when the reasoner CHECK is *sound*, once iteration finishes we have the following specification

$$(\left\|\hat{\chi}^{\mathbf{B}} - \chi^{\mathbf{B}}\right\|_p \leq \epsilon) \wedge (\hat{\chi}^{\Theta \setminus \mathbf{B}} = \chi^{\Theta \setminus \mathbf{B}}) \Rightarrow |\hat{c} - c| \leq \delta. \tag{7}$$

holds on network $f$, where $\hat{\chi}^{\mathbf{B}}$ is the variable representing all the possible assignments of irrelevant features $\mathbf{x}^{\mathbf{B}}$, i.e., $\forall \mathbf{x}^{\mathbf{B}'}$, and the pre-condition $\hat{\chi}^{\Theta \setminus \mathbf{B}} = \chi^{\Theta \setminus \mathbf{B}}$ fixes the values of the explanation features of an instantiated input $\mathbf{x}$. Meanwhile, the post-condition $|\hat{c} - c| \leq \delta$ where $c \mapsto f(\mathbf{x})$ as in Line 3 ensures prediction invariance such that $\delta$ is 0 for classification and otherwise a pre-defined allowable amount of perturbation for regression. To this end, for some specific input $\mathbf{x}$ we have the following property

$$\forall \mathbf{x}^{\mathbf{B}'}. (\left\|\mathbf{x}^{\mathbf{B}'} - \mathbf{x}^{\mathbf{B}}\right\|_p \leq \epsilon) \Rightarrow |f(\mathbf{x}') - f(\mathbf{x})| \leq \delta. \tag{8}$$

holds. Here we prove by construction. According to Equation (1) of Definition 2.1, if the irrelevant features $\mathbf{x}^{\mathbf{B}}$ satisfy the above property, then we call the rest features $\mathbf{x}^{\mathbf{A}}$ a *robust* explanation with respect to network $f$ and input $\mathbf{x}$. □

### A.3 Proof for Theorem 3.4

*Theorem 3.4* (Optimality). If the CHECK sub-procedure is *sound* and *complete*, then the *robust* explanation $\mathbf{x}^\mathbf{A}$ returned by Algorithm 1 is *optimal* – this satisfies Equation (2) of Definition 2.1.

*Proof.* We prove this by contradiction. From Equation (2) of Definition 2.1, we know that explanation $\mathbf{x}^\mathbf{A}$ is optimal if, for any feature $\chi$ in the explanation, there always exists an $\epsilon$-perturbation on $\chi$ and the irrelevant features $\mathbf{x}^\mathbf{B}$ such that the prediction alters. Let us suppose $\mathbf{x}^\mathbf{A}$ is not optimal, then there exists a feature $\chi$ in $\mathbf{x}^\mathbf{A}$ such that no matter how to manipulate this feature $\chi$ into $\chi'$ and the irrelevant features $\mathbf{x}^\mathbf{B}$ into $\mathbf{x}^{\mathbf{B}'}$, the prediction always remains the same. That is,

$$\exists\, \chi \in \mathbf{x}^\mathbf{A}. \,\forall\, \mathbf{x}^{\mathbf{B}'}, \chi'. \left\| (\mathbf{x}^\mathbf{B} \oplus \chi) - (\mathbf{x}^{\mathbf{B}'} \oplus \chi') \right\|_p \leq \epsilon \Rightarrow |f(\mathbf{x}) - f(\mathbf{x}')| \leq \delta, \quad (9)$$

where $\oplus$ denotes concatenation of two features. When we pass this input $\mathbf{x}$ and network $f$ into the VERIX framework, suppose Algorithm 1 examines this feature $\chi$ at the $i$-th iteration, then as in Line 7, the current irrelevant set of indices is $\mathbf{B}^+ \mapsto \mathbf{B} \cup \{i\}$, and accordingly the pre-conditions are

$$\phi \mapsto (\left\| \hat{\chi}^{\mathbf{B} \cup \{i\}} - \chi^{\mathbf{B} \cup \{i\}} \right\|_p \leq \epsilon) \wedge (\hat{\chi}^{\Theta \backslash (\mathbf{B} \cup \{i\})} = \chi^{\Theta \backslash (\mathbf{B} \cup \{i\})}). \quad (10)$$

Because $\hat{\chi}^{\mathbf{B} \cup \{i\}}$ is the variable representing all the possible assignments of irrelevant features $\mathbf{x}^\mathbf{B}$ and the $i$-th feature $\chi$, i.e., $\forall\, \mathbf{x}^{\mathbf{B}'}, \chi'$, and meanwhile

$$\hat{\chi}^{\Theta \backslash (\mathbf{B} \cup \{i\})} = \chi^{\Theta \backslash (\mathbf{B} \cup \{i\})} \quad (11)$$

indicates that the other features are fixed with specific values of this $\mathbf{x}$. Thus, with $c \mapsto f(\mathbf{x})$ in Line 3, we have the specification $\phi \Rightarrow |\hat{c} - c| \leq \delta$ holds on input $\mathbf{x}$ and network $f$. Therefore, if the reasoner CHECK is *sound* and *complete*,

$$\text{CHECK}(f, \phi \Rightarrow |\hat{c} - c| \leq \delta) \quad (12)$$

will always return True. Line 10 assigns True to HOLD, and index $i$ is then put into the irrelevant set $\mathbf{B}$ thus $i$-th feature $\chi$ in the irrelevant features $\mathbf{x}^\mathbf{B}$. However, based on the assumption, feature $\chi$ is in explanation $\mathbf{x}^\mathbf{A}$, so $\chi$ is in $\mathbf{x}^\mathbf{A}$ and $\mathbf{x}^\mathbf{B}$ simultaneously – a contradiction occurs. Therefore, Theorem 3.4 holds. $\qquad\square$

## B Runtime performance and scalability

### B.1 Runtime performance when using sound and complete verifiers

Table 4: Average execution time (seconds) of CHECK and VERIX for *complete* verification. In particular, magnitude $\epsilon$ is set to $3\%$ across the Dense, Dense (large), CNN models and the MNIST, TaxiNet, GTSRB datasets for sensible comparison.

|  | Dense | | Dense (large) | | CNN | |
|---|---|---|---|---|---|---|
|  | CHECK | VERIX | CHECK | VERIX | CHECK | VERIX |
| MNIST ($28 \times 28$) | 0.013 | 160.59 | 0.055 | 615.85 | 0.484 | 4956.91 |
| TaxiNet ($27 \times 54$) | 0.020 | 114.69 | 0.085 | 386.62 | 2.609 | 8814.85 |
| GTSRB ($32 \times 32 \times 3$) | 0.091 | 675.04 | 0.257 | 1829.91 | 1.574 | 12935.27 |

We analyze the empirical time *complexity* of our VERIX approach in Table 4. The model structures are described in Appendix D.4. Typically, the individual pixel checks (CHECK) return a definitive answer (True or False) within a second on dense models and in a few seconds on convolutional networks. For image benchmarks such as MNIST and GTSRB, larger inputs or more complicated models result in longer (pixel- and image-level) execution times for generating explanations. As for TaxiNet as a regression task, while its pixel-level check takes longer than that of MNIST, it is actually faster in total time on dense models because TaxiNet does not need to check against other labels.

### B.2 Scalability on more complex models using sound but *incomplete* analysis

Table 5: Average execution time (seconds) of CHECK and VERIX for *incomplete* verification. Magnitude $\epsilon$ is $3\%$ for both MNIST-sota and GTSRB-sota models.

|  | # ReLU | # MaxPool | CHECK | VERIX |
|---|---|---|---|---|
| MNIST-sota | 50960 | 5632 | 2.31 | 1841.25 |
| GTSRB-sota | 106416 | 5632 | 8.54 | 8770.15 |

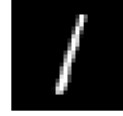 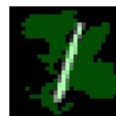 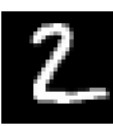 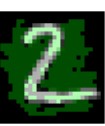 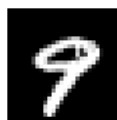 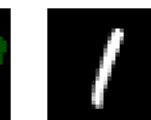 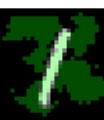

Figure 10: Sound but *incomplete* CHECK procedure DeepPoly contributes to robust but *not optimal* (larger than necessary) VERIX explanations for the convolutional network MNIST-sota.

The *scalability* of VERIX can be improved if we perform incomplete verification, for which we re-emphasize that the soundness of the resulting explanations is not undermined though optimality is no longer guaranteed, i.e., they may be larger than necessary. To illustrate, we deploy the incomplete DeepPoly [46] analysis (implemented in Marabou) to perform the CHECK sub-procedure. Table 5 reports the runtime performance of VERIX when using incomplete verification on state-of-the-art network architectures with hundreds of thousands of neurons. See model structures in Appendix D, Tables 7 and 9. Experiments were performed on a cluster equipped with Intel Xeon E5-2637 v4 CPUs running Ubuntu 16.04. We set a time limit of 300 seconds for each CHECK call. Moreover, in Figure 10, we include some example explanations for the convolutional model MNIST-sota when using the sound but incomplete CHECK procedure DeepPoly. We can see that they indeed appear larger than the optimal explanations when the complete Marabou reasoner is used. We remark that, as soundness of the explanations is not undermined, they still provide guarantees against perturbations on the irrelevant pixels. Interestingly, MNIST explanations on convolutional models tend to be less scattered than these on fully-connected models, as shown in Figures 4b and 6b, due to the effect of convolutions. In general, the scalability of VERIX will grow with that of verification tools, which has improved significantly in the past several years as demonstrated by the results from the Verification of Neural Networks Competitions (VNN-COMP) [3].

# C   Supplementary related work and discussion

## C.1   Related work (cont.)

Continued from Section 5, our work expands on [33] in four important ways: (i) we focus on $\epsilon$-ball perturbations and perception models, whose characteristics and challenges are different from those of NLP models; (ii) whereas [33] simply points to existing work on hitting sets and minimum satisfying assignments for computing optimal robust explanations, we provide a detailed algorithm with several illustrative examples, and include a concrete traversal heuristic that performs well in practice; we believe these details are useful for anyone wanting to produce a working implementation; (iii) we note for the first time the relationship between optimal robust explanations and counterfactuals; and (iv) we provide an extensive evaluation on a variety of perception models. We also note that in some aspects, our work is more limited: in particular, we use a simpler definition of optimal robust explanation (without a cost function) as our algorithm is specialized for the case of finding explanations with the fewest features.

We discuss some further related work in the formal verification community that are somewhat centered around interpretability or computing minimal explanations. [14] uses formal techniques to identify input regions around an adversarial example such that all points in those regions are also guaranteed to be adversarial. Their work improves upon previous work on identifying *empirically robust* adversarial regions, where points in the regions are empirically likely to be adversarial. Analogously, our work improves upon informal explanation techniques like Anchors. [12] is similar to [14] in that it also computes pre-images of neural networks that lead to bad outputs. In contrast, we compute a subset of input features that preserves the neural network output. [67] is more akin to our work, with subtle yet important differences. Their goal is to identify a *minimal* subset of input features that when *corrected*, *changes* a network's prediction. In contrast, our goal is to find a *minimal* subset of input features that when *fixed*, *preserves* a network's prediction. These two goals are related but not equivalent: given a correction set found by [67], a sound but non-minimal VERIX explanation can be obtained by fixing all features not in the correction set along with one of the features in the correction set. Symmetrically, given a VERIX explanation, a sound but non-minimal correction can be obtained by perturbing all the features not in the explanation along with one of the features in the explanation. This relation is analogous to that between minimal correction sets and minimal unsatisfiable cores in constraint satisfaction (e.g., [35]). Both are considered standard explanation strategies in that field.

## C.2   Verification of neural networks

Researchers have investigated how automated reasoning can aid verification of neural networks with respect to formally specified properties [36, 22], by utilizing reasoners based on abstraction [66, 46, 17, 49, 41, 54, 55, 2, 65, 57, 59] and search [15, 30, 31, 23, 56, 48, 21, 7, 13, 44, 62, 61, 5, 32, 16, 58, 60]. Those approaches mainly focus on verifying whether a network satisfies a certain pre-defined property (e.g., robustness), i.e., either prove the property holds or disprove it with a counterexample. However, this does not shed light on *why* a network makes a specific prediction. In this paper, we take a step further, repurposing those verification engines as sub-routines to inspect the decision-making process of a model, thereby explaining its behavior (through the presence or absence of certain input features). The hope is that these explanations can help humans better interpret machine learning models and thus facilitate appropriate deployment.

## C.3   Trustworthiness of the explanations

For a fixed model $f$ and a fixed input $\mathbf{x}$, a *unique* traversal order passed into Algorithm 1 contributes to a *unique* explanation. That said, different traversals can produce different explanations with various sizes. A natural question then is how to evaluate which of these explanations are more *trustworthy* than others? Following a suggestion by an anonymous reviewer, we propose that the *size* of an explanation can be used as a rough proxy for its trustworthiness, since our experiments show that explanations from sensitivity traversals are much smaller and more sensible than those explanations from random traversals. This also suggests that the global cardinality-minimal explanation might be more trustworthy than its locally minimal counterparts. However, as mentioned above, computing a globally minimal explanation is extremely expensive computationally. We view the exploration of better ways to approach globally minimum explanations as a promising direction for future research.

# D Model specifications

Apart from those experimental settings in Section 4, we include detailed model specifications for reproducibility and reference purposes. Although evaluated on the MNIST [34], GTSRB [47], and TaxiNet [29] image datasets – MNIST and GTSRB in classification and TaxiNet in regression, our VERIX framework can be generalized to other machine learning applications such as natural language processing. As for the sub-procedure CHECK of Algorithm 1, while VERIX can potentially incorporate existing automated reasoners, we deploy the neural network verification tool Marabou [31]. While it supports various model formats such as .pb from TensorFlow [1] and .h5 from Keras [9], we employ the cross platform .onnx format for better Python API support. When importing a model with softmax as the final activation function, we remark that, for the problem to be *decidable*, one needs to specify the outputName parameter of the read_onnx function as the pre-softmax logits. As a workaround for this, one can also train the model without softmax in the last layer and instead use the SoftmaxLoss loss function from the tensorflow_ranking package. Either way, VERIX produces consistent results.

## D.1 MNIST

For MNIST, we train a fully-connected feed-forward neural network with 3 dense layers activated with ReLU (first 2 layers) and softmax (last classification layer) functions as in Table 6, achieving 92.26% accuracy. While the MNIST dataset can easily be trained with accuracy as high as 99.99%, we are more interested in whether a very simple model as such can extract sensible explanations – the answer is yes. Meanwhile, we also train several more complicated MNIST models, and observe that their optimal explanations share a common phenomenon such that they are relatively more scattered around the background compared to the other datasets. This cross-model observation indicates that MNIST models need to check both the presence and absence of white pixels to recognize the handwritten digits correctly. Besides, to show the scalability of VERIX, we also deploy incomplete verification on state-of-the-art model structure as in Table 7.

## D.2 GTSRB

As for the GTSRB dataset, since it is not as identically distributed as MNIST, to avoid potential distribution shift, instead of training a model out of the original 43 categories, we focus on the top first 10 categories with highest occurrence in the training set. This allows us to obtain an appropriate model with high accuracy – the convolutional model we train as in Table 8 achieves a test accuracy of 93.83%. It is worth mentioning that, our convolutional model is much more complicated than the simple dense model in [26], which only contains one hidden layer of 15 or 20 neurons trained to distinguish two MNIST digits. Also, as shown in Table 5 of Appendix B, we report results on the state-of-the-art GTSRB classifier in Table 9.

## D.3 TaxiNet

Apart from the classification tasks performed on those standard image recognition benchmarks, our VERIX approach can also tackle regression models, applicable to real-world safety-critical domains. In this vision-based autonomous aircraft taxiing scenario [29] of Figure 9, we train the regression model in Table 10 to produce an estimate of the cross-track distance (in meters) from the ownship to the taxiway centerline. The TaxiNet model has a mean absolute error of 0.824 on the test set, with no activation function in the last output layer.

## D.4 Dense, Dense (large), and CNN

In Appendix B, we analyze execution time of VERIX on three models with increasing complexity: Dense, Dense (large), and CNN as in Tables 11, 12, and 13, respectively. To enable a fair and sensible comparison, those three models are used across the MNIST, TaxiNet, and GTSRB datasets with only necessary adjustments to accommodate each task. For example, in all three models $h \times w \times c$ denotes different input size height $\times$ width $\times$ channel for each dataset. For the activation function of the last layer, softmax is used for MNIST and GTSRB while TaxiNet as a regression task needs no such activation. Finally, TaxiNet deploys he_uniform as the kernel_initializer parameter in the intermediate dense and convolutional layers for task specific reason.

Table 6: Structure for the MNIST classifier.

| Layer Type | Parameter | Activation |
|---|---|---|
| Input | $28 \times 28 \times 1$ | – |
| Flatten | – | – |
| Fully Connected | 10 | ReLU |
| Fully Connected | 10 | ReLU |
| Fully Connected | 10 | softmax |

Table 8: Structure for the GTSRB classifier.

| Type | Parameter | Activation |
|---|---|---|
| Input | $32 \times 32 \times 3$ | – |
| Convolution | $3 \times 3 \times 4$ (1) | – |
| Convolution | $2 \times 2 \times 4$ (2) | – |
| Fully Connected | 20 | ReLU |
| Fully Connected | 10 | softmax |

Table 7: Structure for the MNIST-sota classifier.

| Type | Parameter | Activation |
|---|---|---|
| Input | $28 \times 28 \times 1$ | – |
| Convolution | $3 \times 3 \times 32$ | ReLU |
| Convolution | $3 \times 3 \times 32$ | ReLU |
| MaxPooling | $2 \times 2$ | – |
| Convolution | $3 \times 3 \times 64$ | ReLU |
| Convolution | $3 \times 3 \times 64$ | ReLU |
| MaxPooling | $2 \times 2$ | – |
| Flatten | – | – |
| Fully Connected | 200 | ReLU |
| Dropout | 0.5 | – |
| Fully Connected | 200 | ReLU |
| Fully Connected | 10 | softmax |

Table 9: Structure for the GTSRB-sota classifier.

| Type | Parameter | Activation |
|---|---|---|
| Input | $28 \times 28 \times 1$ | – |
| Convolution | $3 \times 3 \times 32$ | ReLU |
| Convolution | $3 \times 3 \times 32$ | ReLU |
| Convolution | $3 \times 3 \times 64$ | ReLU |
| MaxPooling | $2 \times 2$ | – |
| Convolution | $3 \times 3 \times 64$ | ReLU |
| Convolution | $3 \times 3 \times 64$ | ReLU |
| MaxPooling | $2 \times 2$ | – |
| Flatten | – | – |
| Fully Connected | 200 | ReLU |
| Dropout | 0.5 | – |
| Fully Connected | 200 | ReLU |
| Fully Connected | 10 | softmax |

Table 10: Structure for the TaxiNet model.

| Type | Parameter | Activation |
|---|---|---|
| Input | $27 \times 54 \times 1$ | – |
| Flatten | – | – |
| Fully Connected | 20 | ReLU |
| Fully Connected | 10 | ReLU |
| Fully Connected | 1 | – |

Table 11: Structure for the `Dense` model.

| Layer Type | Parameter | Activation |
|---|---|---|
| Input | $h \times w \times c$ | – |
| Flatten | – | – |
| Fully Connected | 10 | ReLU |
| Fully Connected | 10 | ReLU |
| Fully Connected | 10 / 1 | softmax / – |

Table 12: Structure for `Dense` (`large`).

| Layer Type | Parameter | Activation |
|---|---|---|
| Input | $h \times w \times c$ | – |
| Flatten | – | – |
| Fully Connected | 30 | ReLU |
| Fully Connected | 30 | ReLU |
| Fully Connected | 10 / 1 | softmax / – |

Table 13: Structure for the `CNN` model.

| Layer Type | Parameter | Activation |
|---|---|---|
| Input | $h \times w \times c$ | – |
| Convolution | $3 \times 3 \times 4$ | – |
| Convolution | $3 \times 3 \times 4$ | – |
| Fully Connected | 20 | ReLU |
| Fully Connected | 10 / 1 | softmax / – |

