## B.1  Sensitivity vs. random traversal to generate explanations

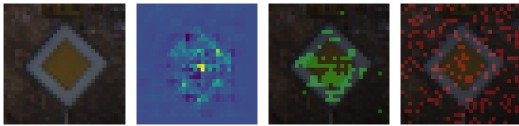

(a) GTSRB "priority road"; sensitivity; explanations from sensitivity (green) and random (red) traversals.

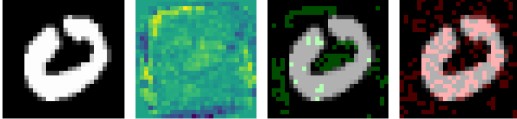

(b) MNIST "0"; sensitivity; explanations from sensitivity (green) and random (red) traversals.

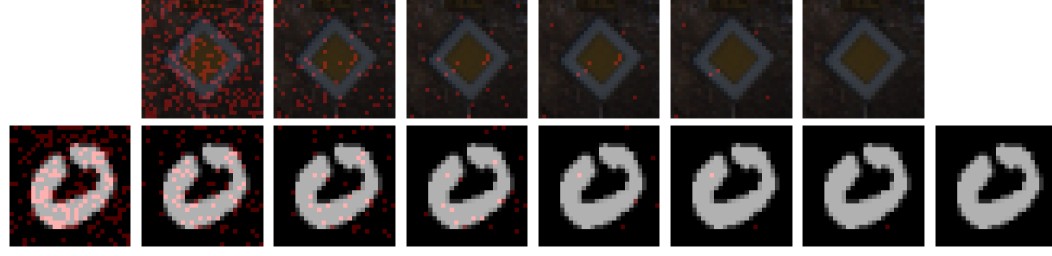

(c) Empty intersection of 6 (top) and 8 (bottom) random explanations for "priority road" and "0", respectively.

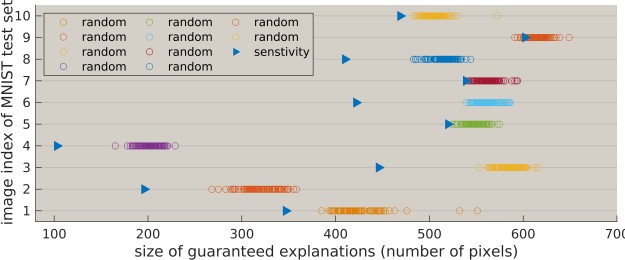

(d) Sensitivity vs. random traversals in explanation size. Each blue triangle denotes 1 deterministic explanation from sensitivity ranking, and each bunch of circles represents 100 explanations from random traversals.

Figure 10: VERIX explanations when using *sensitivity* (green) and random (red) traversals.

To show the advantage of the *sensitivity* traversal, Figure 10 compares VERIX explanations using sensitivity-based and random traversal orders. The first column of Figures 10a and 10b shows the original image; the second a heatmap of the sensitivity (with deletion $\mathcal{T}(\chi) = 0$ for GTSRB and reversal $\mathcal{T}(\chi) = \overline{\chi} - \chi$ for MNIST because deleting background pixels of MNIST images may contribute to little confidence change as they often have zero values); and the third and fourth columns show explanations using the sensitivity and random traversals, respectively. Sensitivity, as shown in the heatmaps, prioritizes pixels that have more influence on the network's prediction. In contrast, a random ranking is simply a shuffling of all the pixels. We observe that the sensitivity traversal generates smaller and more sensible explanations. Furthermore, we also explore the idea of using intersections of explanations generated from random traversals. Specifically, for both images, we randomly traverse all input features 10 times and produce 10 explanations. In Figure 10c, we show the result of the first random explanation, followed by the result of intersecting this explanation with more and more random explanations. The end result is an empty set (last one in each row). This strongly emphasizes the necessity of a sensible traversal, for which we propose the feature-level sensitivity traversal. In Figure 10d, we