# OpenReview forum: "VeriX: Towards Verified Explainability of Deep Neural Networks"
_NeurIPS.cc/2023/Conference — NeurIPS 2023 poster_

### Official Review · Reviewer_oNV3 · 2023-07-03

**Soundness:** 4 excellent
**Presentation:** 4 excellent
**Contribution:** 3 good
**Rating:** 7
**Confidence:** 4

**Summary:**

This paper presents an elegant approach for calculating the optimal robust explanation for a DNN prediction. Intuitively, an explanation is robust if small perturbations of the the features not included in the explanation do not modify the model output. A robust explanation is optimal if small perturbations to any of the features included in the explanation changes the model output. The paper adapts this notion of optimal robust explanation from prior work [31]. The main contribution is the VERIX algorithm for calculating such explanations. This algorithm iterates over each input feature in sequence and uses neural network verification tools as a sub-procedure. Although the algorithm calculates an optimal robust explanation, the explanation is not necessarily the smallest since the model can have multiple optimal robust explanations for a prediction. The size of the calculated explanation crucially depends on the order in which the input features are traversed. The paper presents a heuristic based on input feature sensitivity that seems to work well in practice. For every feature included in the explanation, the algorithm also returns a counterfactual, i.e., an input with a different value for the feature under consideration that would change the model output. The approach is evaluated on models trained for MNIST, GTSRB, and TaxiNet datasets.

**Strengths:**

The paper is very well-written and presents a simple, easy to understand algorithm for computing optimal robust explanations. The authors are upfront about the limitations of the approach which is much appreciated. The empirical evaluations do a good job of addressing different questions of interest such as the effect of $\epsilon$ on the explanations and the effect of feature traversal order.

**Weaknesses:**

My primary concerns about the approach are about its scalability and the usefulness of the calculated explanations.

- The algorithm repeatedly invokes neural network verification tools. Such tools are notoriously hard to scale, limiting the applicability of the approach to large, state-of-the-art networks. It would have been very interesting if the paper included an evaluation using sound but incomplete tools based on techniques like abstract interpretation. Such tools typically demonstrate greater scalability. It would be instructive to see the degradation in the quality of the explanation due to the use of these scalable tools. In the other direction, it would also be instructive to demonstrate the lack of scalability of tools like [4] that calculate the minimum sized explanations.

- My concern about the usefulness of the calculated explanations is a more general concern about this line of work. Although I do not expect the authors to fully resolve this question, it would be useful to have a discussion about this in the paper. For instance, how can developers use these explanations? What specific data points should be picked for computing the explanations? Are there existing studies that show that such explanations are useful in practice?

- Finally, I am also a little concerned by the sensitivity of the algorithm to the order in which the features are traversed. It would be helpful to evaluate a few different strategies for the same. Moreover, if the traversal order can lead to vastly different explanations, how does one trust the generated explanation?

**Questions:**

As mentioned above, I would be interested in the authors thoughts about the following questions:

1. Can the approach be usefully scaled to large models with millions of parameters?

2. How can such explanations be useful to developers? How can developers deploy the algorithm in practice to aid in constructing more trustworthy models?

3. Given the sensitivity of the algorithm to the feature traversal order, how does one trust the explanations? Relatedly, can saliency map be a good heuristic for picking the traversal order?

--------------------
Edits after author response:
Based on author responses to my questions, I have increased my score to 7

**Limitations:**

Yes, the authors clearly discuss the limitations of their approach in the paper.

---

> ### Author Rebuttal · Authors · 2023-08-10
>
> We would like to thank the reviewer for their recognition of our work.
>
> Answers to Questions:
>
> ## 1. Scalability of the approach & Evaluation using other verification tools
>
> Due to space limitation, we address this question with all the details in the **“global” response** to all reviewers at the top of this page. Specifically, in 1.1 of the “global” response we reported the *runtime performance* of our approach, and in 1.2 we used the *sound* but *incomplete* verifier CROWN as the CHECK sub-procedure to scale up to more state-of-the-art models then reported results and exhibited example explanations.
>
> Apart from this, we didn’t compare to these approaches [31,4] that compute the cardinality-minimal explanations because they are too computationally difficult to converge for *large* models, *high-dimensional* inputs, as well as *infinite and continuous* perturbations as in our case. For instance, [31] focuses on the $k$-NN box perturbations in their experimental results, which is essentially discrete as it chooses a finite set of the $k$ closest tokens for each word in a text example.
>
> ## 2. Usefulness of such explanations
>
> We discussed the potential usefulness of our explanations via the two practical applications in the paper, namely the *autonomous aircraft taxiing* scenario in section 4.5 and the *dermatology diagnosis model in healthcare* in section 1 Introduction and section 6 Conclusion.
>
> ### 2.1 Compare trustworthiness of models in aircraft taxiing
>
> For the *autonomous aircraft taxiing* application, while users can train various regression models to obtain the cross-track estimates, our explanations can help users decide which of these models are more **trustworthy**. For instance, if a model produces not-so-bad estimates but its explanations do not make sense, e.g., it does not “look at” the $\mathtt{remote \ line}$ at all or it completely discards the central region - whether there is a $\mathtt{centerline}$ or not, then users should at least take extra precautions to decide whether to trust this model.
>
> ### 2.2 Identify potential bias in medical care
>
> For the *dermatology diagnosis* model, our explanations could help identify potential **bias** if it exists. For instance, if the diagnosis model has an explanation based solely on the patient’s skin tones (“$\mathtt{Fitzpatrick}$ I-II, III-IV, V-VI”), then the model is faulty and possibly biased as it should consider the diseased tissue not the skin type to make a diagnosis.
>
> ## 3. Sensitivity based traversal order & how about saliency map
>
> ### 3.1 How does one trust the explanations?
>
> Ultimately, whether we should trust the explanations depends on how they are aligned with our **human perception**. Ideally, the model should be focusing on the pixels that we would focus on. However, such experiments involving real humans are challenging and require an entirely different experimental setup. As an alternative, we use *explanation size* and *generation time* as the two metrics when evaluating the explanations. As shown in Table 2, our explanations are much smaller than those from Anchors and LIME, but at the cost of taking more time.
>
> ### 3.2 Sensitivity traversal relates robustness
>
> Intuitively, our sensitivity traversal measures how each individual pixel is sensitive to the confidence value of the prediction, and then ranks all pixels from the *least* to the *most* sensitive. We know that those least sensitive pixels tend to be ***more robust*** to perturbations therefore they are more likely to be in the irrelevant set. In other words, Algorithm 1 starts with those least sensitive pixels and puts them into the irrelevant set, and eventually leaves the most sensitive pixels in the explanation set.
>
> To show that our deterministic *feature-level sensitivity* is a good heuristic to generate traversal orders, in section 4.3 and **Appendix B.1** we compare it to *random* traversals. From Figure 5, we can see that explanations from sensitivity traversals make much more sense than those from random traversals. Also, Figure 10(d) shows that the size of explanations from sensitivity traversals is much smaller than that from random traversals.
>
> ### 3.3 How about saliency maps?
>
> That said, *saliency maps* can be a good heuristic for picking the traversal order also, as long as they reflect the above-mentioned correlation with the model's prediction confidence. After all, our sensitivity (Figure 5(a) 2nd column) can be regarded as an example of a saliency map. A possible different strategy is to use the *gradient* of the model with respect to each pixel, and then rank the gradient accordingly from small to large.

---

> > ### Comment · Reviewer_oNV3 · 2023-08-15
> > **Response to rebuttal**
> >
> > Thank you for the detailed response to my questions and for pointing me towards the material in the appendix. I am happy to keep my score but I have two follow-up comments.
> >
> > 1. My question about how does one trust the explanation referred to the fact that for a fixed model $f$ and fixed input $x$, the presented algorithm can generate various explanations depending on the traversal order (as demonstrated by Figures 5 and 10). Intuitively, it would seem that when $f$ predicts $y$ on $x$, there ought to be a unique explanation about the behavior of $f$. To put it differently, should we think of the explanation generated using a random traversal order as a "valid" explanation? If yes, then the explanation in the 4th column of Figure 5(a) is as valid as the explanation in 3rd column of Figure 5(a), and we can either conclude that the model is very badly behaved based on the random traversal explanation or well-behaved based on the sensitivity traversal explanation. Why is the explanation generated by sensitivity-based traversal order considered more trustworthy? I think the answer is that the size of an optimal robust explanation is inversely related to its trustworthiness. So, the cardinality-minimal explanation is the most trustworthy, and one can define an order on the trustworthiness of the explanations as the inverse order of their sizes. I think a discussion on the ordering of explanations would strengthen the paper. Related to this, I have a question -- is the intersection of various explanations generated by random traversal a "valid" explanation? If so, then the empty explanation would be the most trustworthy one (based on the previous argument). But my guess is that the intersection of explanations is not a valid explanation; in fact, I think it is not even an optimal robust explanation as per Definition 2.1 and my suggestion would be to remove Fig 10(c) from the paper because I am not sure what it conveys.
> >
> > 2. I think the results presented in Fig 11 are an important aspect of the empirical evaluation and it would be nice if, in addition to the figure, quantitative results about the size of the explanations when using sound but incomplete verifiers were provided (similar to the *size* column in Table 2).

---

> > > ### Author Response · Authors · 2023-08-20
> > >
> > > We thank Reviewer oNV3 for carefully reading our rebuttal.
> > >
> > > ## 1. *Trustworthiness* of the explanations
> > >
> > > Regarding the first follow-up comment, we agree that the lack of a unique explanation is somehow unsatisfying and also agree that the *size* can be used as a rough proxy for trustworthiness.  We will add a discussion as you suggest.
> > >
> > > Figure 10(c) was actually added in response to a previous reviewer of this paper who was curious about the intersections of explanations from random traversals.  We agree this may be confusing and will *remove* the figure and simply note that (as you correctly surmise) intersections of explanations are *not*, in general, explanations.
> > >
> > > ## 2. Quantitative results when using sound but *incomplete* verification
> > >
> > > Following the reviewer’s suggestion, below we report the quantitative results about the explanations generated when using sound but *incomplete* verifiers.
> > >
> > > ### 2.1 *Complete vs incomplete* verifiers regarding explanation *size* and generation *time*
> > >
> > > Recall that in Table 2 we reported the average size (number of pixels) and generation time (seconds) of explanations for MNIST and GTSRB when using complete verification.
> > >
> > > To compare with *incomplete* verification, and, to make sure it’s a fair comparison, we set the same parameters (e.g., perturbation magnitude $\epsilon$) and produce explanations for the same MNIST and GTSRB models on the same test images, only this time using the sound but incomplete CROWN analysis. We report in the table below the new quantitative results (in bold) for the incomplete case.
> > >
> > >
> > > |       |             |   |   |          | MNIST  |   |   |                |            |   |   |   |   |          | GTSRB  |   |   |                |            |
> > > |-------|-------------|---|---|----------|--------|---|---|----------------|------------|---|---|---|---|----------|--------|---|---|----------------|------------|
> > > |       |             |   |   | complete |        |   |   | **incomplete** |            |   |   |   |   | complete |        |   |   | **incomplete** |            |
> > > |       |             |   |   | size     | time   |   |   | size           | time       |   |   |   |   | size     | time   |   |   | size           | time       |
> > > | VeriX | sensitivity |   |   | 180.6    | 174.77 |   |   | **330.8**      | **106.11** |   |   |   |   |   357.0  | 853.91 |   |   | **385.2**      | **858.39** |
> > > |       | random      |   |   | 294.2    | 157.47 |   |   | **467.4**      | **100.19** |   |   |   |   |   383.5  | 814.18 |   |   | **400.2**      | **816.38** |
> > >
> > > ### 2.2 Trade-off between *complete* and *incomplete* verifiers
> > >
> > > We can see that, in general, explanations are ***larger*** when using incomplete analysis, but have ***shorter*** generation times. In other words, there is a trade-off between *complete* and *incomplete* verifiers with respect to the *size* and *generation time* of explanations.
> > >
> > > This makes sense as the incomplete CROWN analysis deploys bound propagation without complete search, so when the verifier returns $\mathtt{UNKNOWN}$ on a certain feature (Line 10 of Algorithm 1), it is *unknown* whether perturbing this feature (together with the current irrelevant set $\mathbf{B}$) *can* or *cannot* change the model's prediction. Therefore, such features are put into the explanation set $\mathbf{A}$, as only $\epsilon$-robust features are put into the irrelevant set $\mathbf{B}$. This results in *larger* explanations. We re-emphasize that such explanations are no longer locally minimal, but they are still sound as $\epsilon$-perturbations on the irrelevant features will definitely *not* alter the prediction.
> > >
> > > Another observation is that on the MNIST dataset, the discrepancy between explanations (size and time) from *complete* and *incomplete* verifiers is more pronounced than the discrepancy on GTSRB. This is because in Table 2 of the paper we set the perturbation magnitude $\epsilon$ to 5% for MNIST and to 0.5% for GTSRB. So when generating explanations on MNIST,  complete search is often required for a precise answer due to the larger $\epsilon$ value, whereas on GTSRB, bound propagation is often enough due to the small $\epsilon$, and thus complete search is not always needed.
> > >
> > > We will include this further analysis in the final version of the paper.

---

> > > > ### Comment · Reviewer_oNV3 · 2023-08-21
> > > >
> > > > I thank the authors for engaging in this discussion and for all the additional work to address my comments. I assume the authors will be able to add the promised additional discussions to the final version of the paper and I am increasing my score to 7.

---

> > > > > ### Author Response · Authors · 2023-08-21
> > > > >
> > > > > Yes we will add the promised discussion to the final version.

---

> > > > ### Comment · Reviewer_VvcV · 2023-08-21
> > > >
> > > > Regarding the comment of the reviewer oNV3 and the response of the authors, I would underline that intersection of this type of explanation is meaningless (i.e. unsound explanation and not robust),  this might be considered perhaps for feature importance explanations (i.e. features selected by ranking) applied by SHAP for example but not for feature attribution explanations.  However, if the intersection of all possible explanations, which can be exponential in the worst case, is not empty then this will result in one optimal counterfactual explanation.

---

### Official Review · Reviewer_VvcV · 2023-07-05

**Soundness:** 3 good
**Presentation:** 3 good
**Contribution:** 2 fair
**Rating:** 5
**Confidence:** 5

**Summary:**

The paper contributes with proposing a robustness-based method to  compute formal explanations that irreducible (minimal subset of input data, often referred as sufficient explantions) for feedforward neural networks (NNs). Formal robustness checker is used in the experiments but the method allows to use any formal robustness solver to verify existence or not of adversarial examples (AEx) when perturbing a feature input $X^i$ to prove its importance or not for the prediction made by the NN model.

One of the drawback of formal methods in XAI is scalability, and NNs is a good example that shows the limits of automated reasoners. A compromise that might be acceptable for users is to restrict the explanation search to the vicinity of the data input and consider explanations that are sound locally (with some distance/perturbation $\epsilon$) but not in the entire domain. And this paper proposes to compute distance-restricted sufficient explanations using robustness solver as alternative to automated reasoners, so that the input intervals to explore in the model is restricted, which is likely to scale for large NNs.

**Strengths:**

- Computing formal local explanations achieved with applying robustness queries to the model is novel
- The proposed solution offers formal guarantees of soundness and precision of the resulting explanations
in contrast to popular post-hoc XAI approaches like LIME, SHAP, Anchor that are heuristic in their nature
- Moreover, the paper propose a nice trick  for the traversal order of the features which allows in practice to generate smaller
explanations, and might be further exploited to speed up the extraction of the explanations with using other
algorithms  like quickplain to drop quickly group of irrelevant features (instead of a single feature at time with the linear algo 1)

- Experimental results show that explanations computed by VeriX are more succinct than explanations returned by Anchor and LIME for two well-known image benchmarks.

**Weaknesses:**

- The paper rediscovers the so-called deletion-based algorithm, outlined in Algorithm 1, that can be traced at least to:
JOHN W. CHINNECK  "Locating Minimal Infeasible Constraint Sets in Linear Programs" ORSA Journal on Computing1991,  initially used to compute min infeasible set (in over constrained formula) and also used by [24] to computing abductive explanations for NNs.
(In the context of XAI, the insight of the algorithm is to delete features and check if there exists AEx, whilst for minimal infeasible/unsatisfiable sets it relaxes a constraint from the formula and if there is an unsatisfiable core.)

- Section 3.2 ( Theorem 3.3, 3.4 and proposition 3.5) rediscovers what has been established for at least 30 years.
The complexity of algo1/deletion-based algorithm is well known in the literature, it’s a linear search procedure that performs $m$ oracle calls ($m$: number of iterations) and other variants of the algorithm allow to reduce $m$  by doing some pre-processing to refine the initial set x to traverse or exploiting unsatisfiable cores (here AEx) to drop some features and thus save  calls to the NP oracle.

- The paper claims that it establishes for the first time the relationship between explanations and counterfactuals. Unfortunately, it overlooked what already exists in the literature and did not demonstrate clearly this relationship. In fact, explanations and counterfactuals
are dual, and this has been shown in XAI: A. Ignatiev "On Relating Explanations and Adversarial Examples." NeurIPS 2019.

**Questions:**

- Explanations delivered by Algo 1 might be empty sets? similarly for the counterfactuals.  This is problematic for a user, which will always be interested in computing (understand the reason behind a prediction) a counterfactual/explanation for any point of the distribution data.

- can you conclude about the duality between explanations and counterfactuals defined under $\epsilon$-ball ($l_\infty$-norm for robustness)? and how to compute (optimal) counterfactuals from explanations in the same why as algo1 that exploits counterfactuals  or in a efficient way?

- Are counterfactuals  returned by algo 1 optimal/minimal (or smallest)? It seems not to be the case!  Then, an important question raises here: how to compute optimal counterfactuals? and in general case do you think that algo1 can be exploited to devise an algorim for computing optimal/smallest counterfactual?

- can you propose an efficient algorithm to enumerate optimal explanations?

- how can we  compute efficiently a ( single) counterfactual based on robustness checking, and directly given input data without generating  (optimal) explanations ?

**Limitations:**

- The paper failed to show clearly and formally the duality between explanations and counterfactuals
- The outlined algorithm reports the basic version of the linear search algorithm and does not implement possible
optimization/refinement to reduce the total number of oracle calls, for example it does not discuss how can be AEx exploited
to improve the procedure.


Remarks:
- Notion of robust does not really make sense in my opinion, given the definition of an explanation.  In fact, by definition, it is set of feature that are important/relevant for the model prediction, and what remain is not. So no matter the value that other features take the predictions will remain the same, (there is no AEx). Indeed robustness and explanations are tightly related, since that AEx and explanations are related (recent work have pointed out the existing connection between both concepts)  but there’s no sense to caracterize robust explanations following def 2.1 and non-robust explanations. A given subset features  A is either an explanation if perturbing $j\in A$ yield to an AEx, or not one if there still exists an AEx although all features of A are maintained to the values of the input vector.

Besides, recently another form of (robust) counterfactual explanations is introduced in J. Jiang et al "Formalising the Robustness of Counterfactual Explanations for Neural Networks" AAAI23, which is relates faithfulness of the explanation after retraining the same DNN. In this case, the notion of robustness is justified.

- Optimality in algo1 should be precised  that it refers to subset minimality. A confusion might be introduced when using in the text optimal and smallest explanations.   Basically, algo1 returns an irreducible subset and not a minimum (or smallest) set.

---

> ### Author Rebuttal · Authors · 2023-08-03
>
> Responses to Weaknesses:
>
> ## 1&2. Algorithm 1, Theorems 3.3 & 3.4, and Proposition 3.5
>
> We didn’t claim to be the first to discover the so-called deletion-based algorithm. Rather, we explore its use in Algorithm 1 to compute *optimal robust explanations* in the context of formal explainable AI.  We appreciate that the reviewer points out the existence of such an algorithm for computing *minimal infeasible sets* which can be traced back to [1]. We will cite [1] and clarify the correlation in our paper. Meanwhile, we cited [24] multiple times (lines 75, 111, 364, 759) in the paper and, particularly, discussed [24] in section 5 Related work.
>
> Our Theorems 3.3 (Soundness) and 3.4 (Optimality) do rely on the general properties of such deletion-based algorithms, but are also specialized for our specific definitions, so they are not identical.
>
> Proposition 3.5 (Complexity) analyzes the complexity of Algorithm 1 with a particular focus on how *a network $f$* and *a $d$-dimensional input $\mathbf{x}$* would affect the complexity. We felt this would be convenient for the reader.  Certainly the analysis is related to similar algorithms.  We would be happy to clarify that in the final version.
>
> [1] J. Chinneck, E. Dravnieks. “Locating minimal infeasible constraint sets in linear programs”. ORSA Journal on Computing, 1991.
>
> ## 3. Relation of our formal explanations and counterfactuals
>
> The reviewer says [2] relates explanations with counterfactuals, however we emphasize that [2] relates explanations with *adversarial examples* **not** counterfactuals, whereas our work relates *formal* explanations with counterfactuals.
>
> ***While adversarial examples are related to counterfactuals, they are not the same.***  Below we quote from [3] the key difference between these two terms:
> > While the strategies used to find the adversarial examples can be the same adopted to retrieve counterfactuals, the objectives are different. For example, adversarial learning applied to images aims at finding a humanly imperceptible change in the input image that changes the prediction outcome with the idea of fooling the classifier. On the contrary, counterfactual explainers applied to images aim at highlighting significant parts of the image that change the prediction outcome with the idea of explaining the classifier.
>
> In our Figure 4(b), we highlighted those significant parts (yellow) in the explanation (green) for the MNIST digit $\mathtt{1}$ to demonstrate that, if these pixels are allowed to change, they can change the classification into $\mathtt{8}$, $\mathtt{5}$, and $\mathtt{2}$, respectively.
>
> [2] A. Ignatiev. "On relating explanations and adversarial examples". NeurIPS 2019.
>
> [3] R. Guidotti. “Counterfactual explanations and how to find them: literature review and benchmarking”. Data Mining and Knowledge Discovery, 2022.
>
> ---
> Answers to Questions:
>
> ## 1. Already clarified two special cases when computing explanations
>
> In lines 98-105, we specifically clarified *two special cases*. The first case says if the model is **$\epsilon$-robust**, then in order to obtain a valid explanation, a larger $\epsilon$ is required. That’s why, in section 4.2 and Figure 8, we visualized the effect of varying $\epsilon$ on explanations to help users choose an appropriate $\epsilon$ to produce meaningful explanations and counterfactuals depending on their applications.
>
> ## 2&3. Both asked how to compute optimal counterfactuals
>
> From the above text quoted from [3], we know adversarial examples and counterfactuals are not identical. In particular, [3] formalizes a list of desirable properties for counterfactuals: *validity, minimality, similarity, plausibility, discriminative power, actionability, causality, and diversity*, some of which are not typically taken into account by adversarial examples. Therefore, while [2] reveals the duality between explanations and adversarial examples, we must remark that such a straightforward duality between explanations and counterfactuals is not explicit, as it must simultaneously consider these desirable properties.
>
> Meanwhile, as emphasized in lines 124-125, our focus in this paper is to compute *optimal robust explanations* **not** optimal counterfactuals. We do not claim optimality for our counterfactuals. Rather, we observe that these counterfactuals can be computed with no extra cost and can be used to better understand the explanations we generate, as shown in Figure 4(b). If the reviewer is particularly interested in computing optimal counterfactuals, we refer to work [4] in which *actionability* and *plausibility* are considered.
>
> [4] A. Parmentier, T. Vidal. “Optimal counterfactual explanations in tree ensembles.” ICML 2021.
>
> ## 4. Enumerate optimal robust explanations
>
> The point of our novel *feature-level sensitivity traversal* (proposed in section 3.3) is to obtain a good optimal robust explanation without having to enumerate all such explanations, as that is computationally prohibitive. If the reviewer is interested in enumeration, we provide references to [5][6] in which the authors enumerated abductive and contrastive explanations.
>
> [5] A. Ignatiev. “From contrastive to abductive explanations and back again”. AIxIA, 2020.
>
> [6] J. Marques-Silva et al. “Explanations for monotonic classifiers”. ICML 2021.
>
> ## 5. Counterfactuals computed without generating (optimal) explanations
>
> In Algorithm 1, $\mathbf{C}$ denotes a set of counterfactuals and $m$ is a single counterfactual.
>
> Therefore, in the  CHECK sub-procedure (line 10 of Algorithm1), we know that we do obtain a single counterfactual $m$ the *very first time* $\mathtt{HOLD}$ is false, and we efficiently obtain such counterfactuals *every time* when $\mathtt{HOLD}$ is false. In other words, we do not need to wait for the algorithm to finish in order to get counterfactuals.
>
> Due to space limitation, we address the rest of the comments in the **“global” response** (starting from point 2) to all reviewers at the top of the page.

---

> > ### Comment · Reviewer_VvcV · 2023-08-20
> >
> > Thanks for the response and clarifications.
> >
> > A1. That is the issue deletion-based approach was never mentioned in the paper (unless I am missing something) while the main contribution is the deletion-based algorithm using robustness reasoner as oracle to compute explanations for NNs which has been established using other NP-oracles in XAI, and that what ensures the optimality of the explanations. This should be explicitly clarified in the paper.
> >
> > A2. This is trivial, complexity of deletion-based approach is well-known and Algorithm 1 instruments calls to an NP-oracle. Proposition 3.5 overviews the complexity of deletion-based approach. Theorem 3.4 overviews the minimality of the solution computed by deletion-based algorithm which represents the purpose of this method.
> >
> > A3. Even though, the counterfactuals are not optimal then how the algorithm checks actionability and causality? Formal robustness checker considers that any datapoints in some distance p is possible. As results, there might be adversarial examples that are not
> > from the data distribution. Then, the counterfactuals saved in **C**  might not always respects the two properties (suggest changes
> > that cannot happened given the data distribution) and thus they are just adversarial examples, and not explanations.
> > Please clarify this in Section 3 if this is not the case.

---

> > > ### Author Response · Authors · 2023-08-20
> > >
> > > We thank Reviewer VvcV for engaging with the rebuttal and updating their score.
> > >
> > > ## 1&2. Algorithm 1, Theorems 3.3 & 3.4, and Proposition 3.5
> > >
> > > As promised, we will **cite the paper [1] from 1991** and explicitly clarify the correlation between this *deletion-based approach* from [1] and our Algorithm 1.
> > >
> > > We included Proposition 3.5 (Complexity), with a particular focus on a neural network $f$ and a $d$-dimensional input $\mathbf{x}$, for *completeness* as we felt this would be convenient for the reader.  We included Theorems 3.3 (Soundness) and 3.4 (Optimality) for similar reasons.
> > >
> > > However, we agree that these theorems are not difficult and are closely related to the complexity of the deletion-based approach. We will also **clarify this** in the final version.
> > >
> > > [1] J. Chinneck, E. Dravnieks. “Locating minimal infeasible constraint sets in linear programs”. ORSA Journal on Computing, 1991.
> > >
> > > ## 3. Ensuring counterfactuals are from data distribution
> > >
> > > The reviewer’s notes about counterfactuals are interesting.  An interesting direction for future work would indeed be defining and understanding a notion of optimal counterfactuals.
> > >
> > > Regarding ***actionability/feasibility*** ([2]) of counterfactuals.  In most cases, this can be addressed by using an appropriate value for $\epsilon$ in line 8 of Algorithm 1. In particular, a value should be used that ensures  that counterexamples will always be from the input distribution.
> > >
> > > Take *perception networks* as an example. When preprocessing image data, it is common to normalize pixel values into the range of $[0, 1]$, so each pixel is represented by a real value from $0$ to $1$. In other words, each feature $\chi$ of our input $\mathbf{x}$ satisfies $\chi \in [0, 1]$. Let’s say we impose 5%-perturbation, then each $\chi$ can be perturbed within $[\max(0, \chi - 0.05), \min(1, \chi + 0.05)]$. That is, we always make sure all the possible perturbations are within the input data distribution. Consequently, the counterexamples returned by verifiers are always from the data distribution, which makes the counterfactuals ***actionable***.
> > >
> > > An interesting example would be the $\mathtt{age}$ feature in some *tabular data* such as in the scenario of applying for a bank loan, where candidates have various features such as $\mathtt{age}$ and $\mathtt{salary}$. While $\mathtt{salary}$ may increase or decrease, in reality a candidate’s $\mathtt{age}$ cannot decrease. For such features, we would suggest imposing case-by-case perturbations that, for example, allow $\mathtt{salary}$ to increase/decrease in a reasonable range but do not allow the $\mathtt{age}$ feature to change.
> > >
> > > We will **clarify this** in the final version of the paper.
> > >
> > > [2] R. Guidotti. “Counterfactual explanations and how to find them: literature review and benchmarking”. Data Mining and Knowledge Discovery, 2022.

---

> > > > ### Comment · Reviewer_VvcV · 2023-08-21
> > > >
> > > > Thanks for the responses and the reference for 3.
> > > >
> > > > I do not see how the perturbation of image data input within the domain interval of the pixels can guarantee to generate in distribution data. In the other had,  there are techniques in OOD detection for image data that are applying perturbations on training data to generate OOD samples. However, I confess my of expertise in that and let the other reviewers comment on this ...

---

> > > > > ### Author Response · Authors · 2023-08-21
> > > > >
> > > > > Our pixel example assumes that all pixels in the range [0,1] are in the distribution.  If this is not the case, then something more sophisticated is needed.  One possibility is to estimate the distribution by using a GAN trained on the distribution data.  The GAN can then be concatenated with the network to limit the counterfactuals to the (estimated) distribution.  But this is beyond the scope of this paper.

---

### Official Review · Reviewer_zc3P · 2023-07-06

**Soundness:** 4 excellent
**Presentation:** 2 fair
**Contribution:** 2 fair
**Rating:** 5
**Confidence:** 4

**Summary:**

This paper proposes a feature sensitivity evaluation approach to enhance neural network explainability. The proposed approach aims to provide an optimal robust explanation, which is translated as a minimal combination of features that preserves invariance in the network's prediction. Neural network verification tools are leveraged to search for such combination of features.

**Strengths:**

The definition of optimal robust explaination is novel and rigorous. The usage of neural network verification tools not only formally guarantees the soundness of the results but also expands the applicability of those tools.

**Weaknesses:**

1. The approach is apparently computational expensive. Many neural network verifiers are usually slow. Running those verifiers for all pixels in an image will be costly.
2. Many neural network verifiers are sound but barely complete. That means, it is quite possible that HOLD in line 10 of Algorithm 1 is easily False if the parameter \epsilon and \delta are too strict for the verifier.
3. The analysis on the experimental results are not persvasive enough. In line 263, why is it good to include the surrounding pixels as explaining pixels? For instance, in Fig4(a), if a traffic sign has a number '50' in a circle, it must be speed limit sign instead of stopping sign. Then why should we consider the white background important?
4. Question 3 begs a questions on the definition of optimal robust explaination. Since in section 2.1, it is claimed that Definition 2.1 only describes a local optimal explaination. Basically, there is no guarantee on how large and how small the induced sets A can be.

**Questions:**

The questions correspond to the weaknesses.
1. What is the runtime of the experiments? Have the authors compare the runtimes of using different nn verification tools?
2. How strong do \epsilon and \delta influence the results?
3. How different can the global and a local optimal feature set be? Under which conditions, the size difference between those two sets is large and under which conditions the difference is small?

**Limitations:**

The paper It is declared in this paper that the proposed algorithm only produces local optimal solutions. The limitation is not addressed.

---

> ### Author Rebuttal · Authors · 2023-08-10
>
> We would like to thank the reviewer for their recognition of our work.
>
> Answers to Questions:
>
> ## 1. Runtime performance & Comparison to other NN verification tools
>
> Due to space limitation, we address this question with all the details in the **“global” response** to all reviewers at the top of this page. Specifically, in 1.1 of the “global” response we reported the *runtime performance*of our approach, and in 1.2 we used the *sound* but *incomplete* verifier CROWN as the CHECK sub-procedure to scale up to more state-of-the-art models then reported results and exhibited example explanations.
>
> ## 2. Influence of parameters $\epsilon$ and $\delta$
> ### 2.1 Influence of $\epsilon$
>
> The effect of *perturbation magnitude* $\epsilon$ is twofold:
> - if $\epsilon$ is too small, more pixels tend to be robust to such small level of perturbations, therefore $\mathtt{HOLD}$ in line 10 of Algorithm 1 is more likely to be $\mathtt{False}$, and thus more pixels will be in the irrelevant set, resulting in smaller explanations. A special case mentioned in lines 98-103 is that when the input is $\epsilon$-robust, then all the pixels become irrelevant, and thus no valid explanation is produced.
> - if $\epsilon$ is too large,  then more perturbations are allowed in each pixel, and thus fewer pixels are needed to alter the model prediction, i.e., smaller irrelevant sets are produced and therefore larger explanations. The corresponding special case is that if perturbing any feature in an input can change the prediction, then the entire input is an explanation (lines 103-105).
>
> In reality, how strong $\epsilon$ influences the explanations really depends on how **robust** the actual model is on the dataset. Intuitively, if a model is more robust, it requires perturbations in more pixels to change prediction, i.e., larger irrelevant sets and smaller explanation sets. Therefore, to prevent the explanation from being way too small, it makes sense to choose a relatively loose $\epsilon$ to allow more perturbations in each pixel.
>
> When doing the experiments, we trained various models on MNIST and GTSRB, and noticed that MNIST models tend to be *more robust* than GTSRB models since MNIST images are more independent and identically distributed. Therefore, to obtain sensible explanations, it makes more sense to set a *relatively loose* $\epsilon$ for MNIST and a *relatively tight* $\epsilon$ for GTSRB. For instance, in Table 2’s caption, we mentioned that $\epsilon$ is set to $5$% for MNIST and $0.5$% for GTSRB, as these MNIST models are *already robust* to $1$%-perturbations (line 292).
>
> To pick a suitable $\epsilon$ for MNIST, in section 4.2 (Figure 8), we visualized the effect of varying $\epsilon$ on explanations. The users then can use the same method to pick a suitable $\epsilon$ depending on their applications.
>
> ### 2.2 Influence of $\delta$
>
> The output *discrepancy* $\delta$ is used to accommodate both classification (set $\delta = 0$, and thus $f(\mathbf{x}) = f(\mathbf{x}’)$, meaning the class is not changed) and regression ($\delta$ could be some predefined hyper-parameter quantifying the allowable output change), as in lines 82-84.
>
> For classification, recall $f(\mathbf{x}) = \arg \max f_c(\mathbf{x})$, where $f_c(\mathbf{x})$ denotes the confidence value (pre- or post softmax) of classifying as class $c$. As $\delta$ is $0$, it is the **confidence value** $f_c(\mathbf{x})$ that plays a key role. Intuitively, when a prediction has higher confidence, its confidence needs to drop more in order to alter the prediction, which potentially requires perturbations in more pixels, therefore more pixels in the irrelevant set and consequently smaller explanations. For instance, MNIST models tend to have relatively high confidence of classifying an input as a certain digit - higher than that of GTSRB models classifying a traffic sign, so MNIST models often have smaller explanations than GTSRB models, as observed in Table 2.
>
> For regression tasks, recall that $f(\mathbf{x}) = c$, where $c$ is a single quantity, e.g., the cross-track estimate of the autonomous aircraft taxiing scenario in section 4.5. If $\delta$ is small, i.e., $c$ is allowed to change little. Then, given the same perturbation magnitude $\epsilon$, fewer pixels are allowed with such perturbations, therefore resulting in smaller irrelevant sets and thus larger explanations.
>
> ## 3. Clarifying why those surrounding pixels are in the explanation for Fig 4(a) $\mathtt{50 \ mph}$
>
> We would like to remark that it is not “us” who consider the white background in $\mathtt{50 \  mph}$ important; rather, it is “the model” that considers the white background important.
>
> We can speculate as to why this is.  One thing to keep in mind is that the trained GTSRB model is ***not perfect***. A perfect model would be able to correctly recognize all traffic signs in the dataset (accuracy 100%), and, more importantly, we would hope that there are no unnecessary artifacts in its explanations.
>
> Real models are not perfect.  So in this case, we speculate that this GTSRB model checks a few surrounding pixels to “know” that they are the background and are therefore not important for decision-making. And simultaneously, it also checks key pixels in the central region of the image to “know” there is a number $\mathtt{50}$, and therefore be able to predict it as a $\mathtt{50 \ mph}$ road sign.
>
> ## 4. Regarding local and global optimal explanations
>
> In Definition 2.1, an explanation is *locally optimal* in the sense that no proper subset of the explanation is itself an explanation.  We believe that this definition is a good compromise between *computational difficulty* and *usability*.  While a globally optimal explanation might be a bit smaller, it would be much more difficult to compute.  In practice, we don’t expect the globally optimal explanation to be much smaller than the one computed using the sensitivity traversal, which already shows promising results experimentally.

---

### Official Review · Reviewer_Yjtz · 2023-07-12

**Soundness:** 3 good
**Presentation:** 3 good
**Contribution:** 3 good
**Rating:** 7
**Confidence:** 3

**Summary:**

The paper presents a system called VERIX for producing robust explanations and generating counterfactuals for deep neural network decisions. VERIX addresses the problem of explainability in AI by applying constraint solving techniques and a sensitivity-based heuristic to iteratively create explanations and counterfactuals with formal guarantees. This is particularly valuable in high-risk domains like autonomous driving and healthcare. The paper provides a formal definition of an optimal robust explanation and discusses how the system deals with two types of bounded perturbations. A key novelty is the system's ability to compute counterfactuals automatically at no additional cost. The efficacy of the system is demonstrated through several experiments and comparisons with existing methodologies, including an application in an autonomous aircraft taxiing scenario.


**Strengths:**

The paper presents an original and well-motivated approach to producing robust explanations for machine learning models. It integrates constraint solving and sensitivity-based heuristic to generate explanations and counterfactuals that have formal guarantees. A significant strength lies in its applicability to safety-critical domains, where the understanding of model behavior is paramount. Moreover, the ability to generate counterfactuals at no additional computational cost provides a practical advantage. Finally, the detailed comparisons with other explainability methods provide compelling evidence of the superiority of VERIX in terms of its precision and the quality of explanations it provides.


**Weaknesses:**

While the paper provides a thorough evaluation of VERIX, the generalizability of the results could be questioned, as the experimental validation is primarily performed on image datasets. Exploring the performance of VERIX on diverse types of data, including structured and time-series data, could provide a more comprehensive view of the system's applicability. Additionally, the complexity and computational cost of the VERIX algorithm might limit its use for large-scale, high-dimensional data.


**Questions:**

- Could you provide more details on the scalability of VERIX, especially for larger and more complex models?
- How does VERIX perform on non-image data? Were any experiments conducted using such data?
- Could you discuss the potential trade-offs in using VERIX, especially regarding computational cost and complexity?

**Limitations:**

The authors discuss potential limitations of VERIX, including scalability and the computational cost of computing optimal robust explanations. These limitations are adequately addressed. The authors also consider future work, such as extending VERIX to other properties like fairness. However, the potential negative societal impact of the work, if any, is not explicitly addressed.

---

> ### Author Rebuttal · Authors · 2023-08-10
>
> We would like to thank the reviewer for their recognition of our work.
>
> Answers to Questions:
>
> ## 1. Scalability of VeriX for more complex models
>
> Due to space limitation, we address this question with all the details in the **“global” response** to all reviewers at the top of this page. Specifically, in 1.1 of the “global” response we reported the *runtime performance* of our approach, and in 1.2 we scaled up to more state-of-the-art models using the *sound* but *incomplete* verifier CROWN as the CHECK sub-procedure then reported results and exhibited example explanations.
>
> ## 2. Performance of VeriX on non-image data
>
> Yes, our approach can be generalized to non-image data such as tabular data or texts in natural language processing.
>
> ### 2.1 How our method can work on NLP datasets
>
> Here we show how it works on the SST (Stanford Sentiment Treebank) dataset for ***sentiment analysis***. Specifically, we train a neural network on the SST dataset to predict whether the sentiment of a text is *positive* or *negative*. We attach two figures in the PDF of the "global" response to demonstrate our experimental results.
>
> We first perform tokenization and post-padding on the text and convert all the words into integer values. Then we train a model with an embedding layer to transform the integers into floats, i.e., word embeddings. The structure of the model is similar to the MNIST model we trained in the paper, only this time the activation function of the last layer is *sigmoid* rather than ReLU due to sentiment analysis. That is, if the model output is greater than 0.5, the text is positive; otherwise it's negative.
>
> ### 2.2 Get feature-level sensitivity traversal
>
> We take the following example text from the dataset, whose sentiment is *positive*.
>
> $\mathtt{one \ of \  the \ more \ intelligent \ children's \ movies \ to \ hit \ theaters \ this \ year}$
>
> By using the feature-level *sensitivity* traversal (proposed in section 3.3), we plot the sensitivity as the heatmap in Figure 1 of the "global" response. We can see that the zero paddings ("#") have flat sensitivity, which makes sense. The actual words have various sensitivity values, represented by the lighter and deeper colors. Intuitively, this shows how sensitive each word is to the final prediction of the sentiment. For instance, here the word $\mathtt{intelligent}$ is the most sensitive compared to others.
>
> ### 2.3 Obtain our explanations
>
> Once we have the sensitivity traversal, we pass it to Algorithm 1 to compute the explanations.
>
> In Figure 2, we show our VeriX explanations - $\mathtt{intelligent}$ and $\mathtt{movies}$ - highlighted in red. The perturbation magnitude $\epsilon$ is set to 0.015 at the embedding space. Our results show that *any possible* perturbations within 0.015 on the irrelevant tokens (in green) will never flip the model's decision from positive to negative.
>
> ## 3. Trade-offs of VeriX regarding computational cost and complexity
>
> ### 3.1 Complexity of VeriX
>
> As shown in *Proposition 3.5 (Complexity)* in lines 235-237,  the complexity of VeriX depends on (1) the dimension $d$ of input, and (2) $P(f)$, which is the cost of checking a specification over the model $f$.
>
> ### 3.2 Input dimensions $d$
>
> For dimension $d$, we performed computational cost analysis on different image benchmarks (MNIST, TaxiNet, and GTSRB) with various dimensions, as shown in Tables 3 and 4. Take MNIST and GTSRB as an example, when $d$ increases from 28x28 to 32x32x3, we observe that both $\mathtt{CHECK}$ and VeriX times increase for all the models in Tables 3 and 4. That is, when the input dimension increases, the computational cost increases.
>
> ### 3.3 Cost of checking a specification on model $f$
>
> As for $P(f)$, when a model is more complex, it takes longer to check a specification on it. From Table 3, we see that for the same MNIST dataset, when model complexity increases from Dense to Dense (large) and to CNN, the $\mathtt{CHECK}$ time increases from 0.013 to 0.055 and to 0.484. Similarly for TaxiNet and GTSRB.
>
> ### 3.4 Trade-off
>
> Therefore, the ***trade-off*** between computational cost and complexity is that, in order to generate formal explanations for more complex inputs and models, i.e., greater $d$ and $P(f)$, we will need to take into account higher computational cost, e.g., longer execution time, which can be alleviated by allocating more computational resources, i.e., to reduce $P(f)$.  For even more scalability, we can use incomplete methods at the cost of larger explanations.
>
> ---
>
> Responses to Limitations:
>
> ## Potential societal impact
>
> While we explicitly discussed the limitations of our work, we didn’t address the societal impact. Formal methods are designed to *mitigate* negative social impact from incorrect or non-robust models.  Thus, our work aims to bring *positive* societal impact. For instance, by analyzing our formal explanations, users can choose those trustworthy models with sensible explanations better aligned with human perception, thereby facilitating their safe deployment in safety-critical applications such as the autonomous aircraft taxiing scenario in our section 4.5. Moreover, a possible future direction is to extend our method to discover potential bias in medical diagnosis models, and thus to promote fairness in healthcare.

---

### Author Rebuttal · Authors · 2023-08-10

We would like to thank all reviewers for their comments and suggestions.

Due to space limitations, below in **1** we address the common questions from more than one reviewer about the *runtime performance and scalability* of our approach, in **2** we respond to a crucial misunderstanding from Reviewer VvcV regarding the *size* of our formal explanations, and in **3**, we address the “Remarks” section of Reviewer VvcV’s review.

## 1. Runtime performance and scalability of our approach

### 1.1 Runtime performance of the experiments

We already included the analysis of *runtime performance* in **Appendix B.2 (page 18)**.  It is referred to in line 360 of the main paper.

In Table 3, we reported the average execution time of the CHECK sub-procedure and the VeriX algorithm on image benchmarks (MNIST, TaxiNet, GTSRB) for models with increasing complexity - $\mathtt{Dense / Dense (large) / CNN}$. Typically, the CHECK sub-procedure returns a definitive answer (True/False) within a second on these dense models and in a few seconds on these CNN models. The total time to obtain a VeriX explanation varies from hundreds of seconds (for dense models) to thousands of seconds (for CNN models).

As analyzed in Proposition 3.5 (Complexity), the complexity of VeriX depends on (1) the input dimension $d$ and (2) the cost of checking a specification for model $f$, i.e., $P(f)$. Therefore, when the input dimension increases, e.g., from MNIST 28x28 to GTSRB 32x32x3, the CHECK and VeriX times increase accordingly. Meanwhile, for each dataset, when the model complexity increases, e.g., from $\mathtt{Dense}$ to $\mathtt{Dense (large)}$, the cost of checking a specification $P(f)$ increases, and thus the CHECK and VeriX times increase too.

### 1.2  Scalable to more complex models using sound and incomplete verifier

Apart from the sound and complete tool Marabou, we also deployed the sound but *incomplete* **CROWN** analysis to perform the CHECK sub-procedure in order to scale up to more complex models. We should emphasize that, when the deployed verifier is sound but incomplete, the resulting explanations are still *sound* (i.e., provide guarantees against perturbations on the irrelevant pixels) though no longer *optimal* (i.e., no longer subset-minimal so may be larger than necessary).

In Table 4, we reported runtime performance of CHECK and VeriX on MNIST and GTSRB models with hundreds of thousands of neurons (e.g., 106416 ReLUs in $\mathtt{GTSRB}$-$\mathtt{sota}$) using the sound but *incomplete* CROWN analysis as the CHECK sub-procedure. We observe that CHECK returns in seconds and VeriX finishes in thousands of seconds. Moreover, for such *incomplete* verification, in Figure 11 we provided some example explanations for the CNN model $\mathtt{MNIST}$-$\mathtt{sota}$ (model structure in Table 6, page 21). We can see that they tend to be larger than the optimal explanations when using Marabou.

Finally, we would like to remark that the *scalability* of VeriX will grow with that of verification tools, which has improved significantly in the past several years as demonstrated by the results from the Verification of Neural Networks Competitions (VNN-COMP) [1].

[1] S. Bak et al. “The 4th International Verification of Neural Networks Competition (VNN-COMP).” 2023.


## 2. Our explanations are *smaller not larger* than Anchors!

**Reviewer VvcV *mistakenly* claims that our “formal explanations are generally quite large (compared for example to Anchor).”**

We quote what the reviewer said below:
> Formal explanations are generally quite large (compared for example to Anchor), which is likely the case for NNs trained on image data. For example, explanations that represent 40% or 50% of features (~400 pixels in MNIST) might be hard for practitioners to be interpreted/understood. Is there any perspectives in this line to overcome the issue of size for formal explanations?

We are unsure what gave the reviewer this impression, but it is not correct.  On the contrary, our explanations are in fact ***significantly smaller*** than those produced by either LIME or Anchors, as shown in Table 2 (page 8) under the *size* column: our size `180.6` vs Anchor size `494.9` vs LIME size `432.8` for MNIST, and our size `357.0` vs Anchor size `557.7` vs LIME size `452.9` for GTSRB. We are guessing that when Reviewer VvcV specifically mentions “(~400 pixels in MNIST)”, they thought that was our result, but in fact that’s the Anchor result.

So, a key contribution is that our explanations are both *smaller* and provide *formal guarantees*.

- - -
## 3. Responses to Reviewer VvcV’s Remarks

### 3.1 Notion of optimal robust explanations

Reviewer VvcV complains about the use of “*robust*” when talking about explanations.  They say,
> In fact, by definition, it is set of feature that are important/relevant for the model prediction, and what remain is not. So no matter the value that other features take the predictions will remain the same, (there is no AEx).

If, in fact, this were the definition used in the literature, the reviewer would be correct.  But it is not.  Indeed, prominent techniques for producing “*explanations*” (such as LIME and Anchors) do not have the property that there is no adversarial example.  Thus, the introduction of the word “*robust*” seems appropriate to distinguish our definition from theirs.  We would also like to point out that the name “*optimal robust explanation*” is not something we invented, but comes from [31].  We use the same name to emphasize the link to that work and to avoid creating confusion by introducing many names for the same thing.

### 3.2 Optimality already clarified in the paper

In lines 106-115, we do in fact clarify that our definition of optimality is *local* in the sense that it computes a *minimal* subset of features rather than a “minimum (global smallest)” set of features.

---

> ### Comment · Reviewer_VvcV · 2023-08-20
>
> A2. Thanks for pointing out that, indeed Fig 6 shows that for both datasets MNIST and GTSRB, Anchor and LIME explanations
> are larger than explanations provided by VeriX.
>
> A3. In opposition to the notion of explanation of model-agnostic approaches like LIME, SHAP, Anchor, .... it makes sense to highlight the characteristic of robustness. My point is that explanations obtained with formal methods (automatic reasoning) are inherently robust/rigorous, often referred as sufficient explanations. And OR explanations  proposed in this work and [31] are distance-restricted sufficient explanations.

---

> > ### Author Response · Authors · 2023-08-20
> >
> > We thank Reviewer VvcV for engaging with  the rebuttal and updating their score.
> >
> > ## A3. Distance-restricted sufficient explanations
> > Regarding A3, we agree that **distance-restricted sufficient explanation** would be another way to characterize these explanations that is consistent with existing literature.
> >
> > In fact, that's why in lines 75-84 of the paper we mentioned that our notion of optimal robust explanations [31] builds on existing work such as *abductive explanations* [24], *prime implicants* [43], and ***sufficient reasons*** [10].
> >
> >  We will make this connection more explicit in the final version to further clarify things for readers.

---

### Decision · Program_Chairs · 2023-09-21

**Decision:**

Accept (poster)

**Comment:**

The authors develop a framework for computing explanations for ML predictions in terms of feature attributions with formal robustness guarantees. The authors introduce several novel ideas and compelling experiments demonstrating the power of their approach on image classification and tabular domains. While the reviewers identified and raised some issues with the paper initially, these were successfully addressed in my opinion during the rebuttal phase. Hence I recommend acceptance.